# Shallow and Deep Networks are Near-Optimal Approximators of Korobov Functions

**Moïse Blanchard**
Operations Research Center
Massachusetts Institute of Technology
Cambridge, MA, 02139
moiseb@mit.edu

**Amine Bennouna**
Operations Research Center
Massachusetts Institute of Technology
Cambridge, MA, 02139
amineben@mit.edu

## Abstract

In this paper, we analyze the number of neurons and training parameters that a neural network needs to approximate multivariate functions of bounded second mixed derivatives — Korobov functions. We prove upper bounds on these quantities for shallow and deep neural networks, drastically lessening the curse of dimensionality. Our bounds hold for general activation functions, including ReLU. We further prove that these bounds nearly match the minimal number of parameters any continuous function approximator needs to approximate Korobov functions, showing that neural networks are near-optimal function approximators.

## 1 Introduction

Neural networks have known tremendous success in many applications such as computer vision and pattern detection (Krizhevsky et al., 2017; Silver et al., 2016). A natural question is how to explain their practical success theoretically. Neural networks are shown to be universal (Hornik et al., 1989): any Borel-measurable function can be approximated arbitrarily well by a neural network with sufficient number of neurons. Furthermore, universality holds for as low as 1-hidden-layer neural network with reasonable activation functions. However, these results do not specify the needed number of neurons and parameters to train. If these numbers are unreasonably high, the universality of neural networks would not explain their practical success.

We are interested in evaluating the number of neurons and training parameters needed to approximate a given function within $\epsilon$ with a neural network. An interesting question is how do these numbers scale with $\epsilon$ and the dimensionality of the problem, i.e., the number of variables. Mhaskar (1996) showed that any function of the Sobolev space of order $r$ and dimension $d$ can be approximated within $\epsilon$ with a 1-layer neural network with $\mathcal{O}(\epsilon^{-\frac{d}{r}})$ neurons and an infinitely differentiable activation function. This bound exhibits the curse of dimensionality: the number of neurons needed for an $\epsilon-$approximation scales exponentially in the dimension of the problem $d$. Thus, Mhaskar's bound raises the question of whether this curse is inherent to neural networks.

Towards answering this question, DeVore et al. (1989) proved that any continuous function approximator (see Section 5) that approximates all Sobolev functions of order $r$ and dimension $d$ within $\epsilon$, needs at least $\Theta(\epsilon^{-\frac{d}{r}})$ parameters. This result meets Mhaskar's bound and confirms that neural networks cannot escape the curse of dimensionality for the Sobolev space. A main question is then for which set of functions can neural networks break this curse of dimensionality.

One way to circumvent the curse of dimensionality is to restrict considerably the considered space of functions and focus on specific structures adapted to neural networks. For example, Mhaskar et al. (2016) showed that compositional functions with regularity $r$ can be approximated within $\epsilon$ with deep neural networks with $\mathcal{O}(d \cdot \epsilon^{-\frac{2}{r}})$ neurons. Other structural constraints have been considered for compositions of functions (Kohler & Krzyżak, 2016), piecewise smooth functions (Petersen & Voigtlaender, 2018; Imaizumi & Fukumizu, 2019), or structures on the data space, e.g., lying on a manifold (Mhaskar, 2010; Nakada & Imaizumi, 2019; Schmidt-Hieber, 2019). Approximation bounds have also been obtained for the function approximation from data under smoothness constraints (Kohler & Krzyżak, 2005; Kohler & Mehnert, 2011) and specifically on mixed smooth Besov

spaces which are known to circumvent the curse of dimensionality (Suzuki, 2018). Another example is the class of Sobolev functions of order $d/\alpha$ and dimension $d$ for which Mhaskar's bound becomes $\mathcal{O}(\epsilon^{-\alpha})$. Recently, Montanelli et al. (2019) considered bandlimited functions and showed that they can be approximated within $\epsilon$ by deep networks with depth $\mathcal{O}((\log \frac{1}{\epsilon})^2)$ and $\mathcal{O}(\epsilon^{-2}(\log \frac{1}{\epsilon})^2)$ neurons. Weinan et al. (2019) showed that the closure of the space of 2-layer neural networks with specific regularity (namely a restriction on the size of the network's weights) is the Barron space. They further show that Barron functions can be approximated within $\epsilon$ with 2-layer networks with $\mathcal{O}(\epsilon^{-2})$ neurons. Similar line of work restrict the function space with spectral conditions, to write functions as limits of shallow networks (Barron, 1994; Klusowski & Barron, 2016; 2018).

In this work, we are interested in more general and generic spaces of functions. Our space of interest is the space of multivariate functions of bounded second mixed derivatives, the Korobov space. This space is included in the Sobolev space but is reasonably large and general. The Korobov space presents two motivations. First, it is a natural candidate for a large and general space included in the Sobolev space where numerical approximation methods can overcome the curse of dimensionality to some extent (see Section 2.1). Second, Korobov spaces are practically useful for solving partial differential equations (Korobov, 1959) and have been used for high-dimensional function approximation (Zenger & Hackbusch, 1991; Zenger, 1991). Recently, Montanelli & Du (2019) showed that deep neural networks with depth $\mathcal{O}(\log \frac{1}{\epsilon})$ and $\mathcal{O}(\epsilon^{-\frac{1}{2}}(\log \frac{1}{\epsilon})^{\frac{3(d-1)}{2}+1})$ neurons can approximate Korobov functions within $\epsilon$, lessening the curse of dimensionality for deep neural networks asymptotically in $\epsilon$. While they used deep structures to prove their result, the question of whether shallow neural networks also break the curse of dimensionality for the Korobov space remains open.

In this paper, we study deep and shallow neural network's approximation power for the Korobov space and make the following contributions:

**1. Representation power of shallow neural networks.** We prove that any Korobov function can be approximated within $\epsilon$ with a 2-layer neural network with ReLU activation function with $\mathcal{O}(\epsilon^{-1}(\log \frac{1}{\epsilon})^{\frac{3(d-1)}{2}+1})$ neurons and $\mathcal{O}(\epsilon^{-\frac{1}{2}}(\log \frac{1}{\epsilon})^{\frac{3(d-1)}{2}})$ training parameters (Theorem 3.1). We further extend this result to a large class of commonly used activation functions (Theorem 3.4). Asymptotically in $\epsilon$, our bound can be written as $\mathcal{O}(\epsilon^{-1-\delta})$ for all $\delta > 0$, and in that sense breaks the curse of dimensionality for shallow neural networks.

**2. Representation power of deep neural networks.** We show that any function of the Korobov space can be approximated within $\epsilon$ with a deep neural network of depth $\lceil \log_2(d) \rceil + 1$ independent of $\epsilon$, with non-linear $\mathcal{C}^2$ activation function, $\mathcal{O}(\epsilon^{-\frac{1}{2}}(\log \frac{1}{\epsilon})^{\frac{3(d-1)}{2}})$ neurons and $\mathcal{O}(\epsilon^{-\frac{1}{2}}(\log \frac{1}{\epsilon})^{\frac{3(d-1)}{2}})$ training parameters (Theorem 4.1). This result improves that of Montanelli & Du (2019) who constructed an approximating neural network with larger depth $\mathcal{O}(\log \frac{1}{\epsilon} \log d)$ –increasing with $\epsilon$– and larger number of neurons $\mathcal{O}(\epsilon^{-\frac{1}{2}}(\log \frac{1}{\epsilon})^{\frac{3(d-1)}{2}+1})$. However, they used ReLU activation function.

**3. Near-optimality of neural networks as function approximators.** Under the continuous function approximators model introduced by DeVore et al. (1989), we prove that any continuous function approximator needs $\Theta(\epsilon^{-\frac{1}{2}}(\log \frac{1}{\epsilon})^{\frac{d-1}{2}})$ parameters to approximate Korobov functions within $\epsilon$ (Theorem 5.2). This lower bound nearly matches our established upper bounds on the number of training parameters needed by deep and shallow neural networks to approximate functions of the Korobov space, proving that they are near-optimal function approximators of the Korobov space.

Table 1 summarizes our new bounds and existing bounds on shallow and deep neural network approximation power for the Korobov space, Sobolev space and bandlimited functions. Our proofs are constructive and give explicit structures to construct such neural networks with ReLU and general activation functions. Our constructions rely on sparse grid approximation introduced by Zenger (1991), and studied in detail in Bungartz (1992); Bungartz & Griebel (2004). Specifically, we use the sparse grid approach to approximate smooth functions with sums of products then construct neural networks which approximate this structure. A key difficulty is to approximate the product function. In particular in the case of shallow neural networks, we propose, to the best of our knowledge, the first architecture approximating the product function with polynomial number of neurons. To derive our lower bound on the number of parameters needed to approximate the Korobov space, we construct a linear subspace of the Korobov space, with large Bernstein width. This subspace is then used to apply a general lower bound on nonlinear approximation derived by DeVore et al. (1989).

The rest of the paper is structured as follows. In Section 2, we formalize our objective and introduce the sparse grids approach. In Section 3 (resp. 4), we prove our bounds on the number of neurons and training parameters for Korobov functions approximation with shallow (resp. deep) networks. Finally, we formalize in Section 5 the notion of optimal continuous function approximators and prove our novel near-optimality result.

| Space | Nb. of neurons | Nb. of training parameters | Depth | Activation $\sigma$ | Reference |
|---|---|---|---|---|---|
| $W^{r,p}$ | $\epsilon^{-\frac{d}{r}}$ | $\epsilon^{-\frac{d}{r}}$ | 1 | $\mathcal{C}^\infty$, non-poly | Mhaskar (1996) |
| $X^{2,\infty}$ | $\epsilon^{-1}(\log\frac{1}{\epsilon})^{\frac{3(d-1)}{2}+1}$ | $\epsilon^{-\frac{1}{2}}(\log\frac{1}{\epsilon})^{\frac{3(d-1)}{2}}$ | 2 | ReLU-like | **This paper** |
| $X^{2,\infty}$ | $\epsilon^{-\frac{3}{2}}(\log\frac{1}{\epsilon})^{\frac{3(d-1)}{2}}$ | $\epsilon^{-\frac{1}{2}}(\log\frac{1}{\epsilon})^{\frac{3(d-1)}{2}}$ | 2 | Sigmoid-like | **This paper** |
| $W^{r,p}$ | $\epsilon^{-\frac{d}{r}}\log\frac{1}{\epsilon}$ | $\epsilon^{-\frac{d}{r}}\log\frac{1}{\epsilon}$ | $\mathcal{O}(\log\frac{1}{\epsilon})$ | ReLU | Yarotsky (2017) Liang & Srikant (2016) |
| Bandlimited functions | $\epsilon^{-2}(\log\frac{1}{\epsilon})^2$ | $\epsilon^{-2}(\log\frac{1}{\epsilon})^2$ | $\mathcal{O}((\log\frac{1}{\epsilon})^2)$ | ReLU | Montanelli et al. (2019) |
| $X^{2,\infty}$ | $\epsilon^{-\frac{1}{2}}(\log\frac{1}{\epsilon})^{\frac{3(d-1)}{2}+1}$ | $\epsilon^{-\frac{1}{2}}(\log\frac{1}{\epsilon})^{\frac{3(d-1)}{2}}$ | $\mathcal{O}(\log d \cdot \log\frac{1}{\epsilon})$ | ReLU | Montanelli & Du (2019) |
| $X^{2,\infty}$ | $\epsilon^{-\frac{1}{2}}(\log\frac{1}{\epsilon})^{\frac{3(d-1)}{2}}$ | $\epsilon^{-\frac{1}{2}}(\log\frac{1}{\epsilon})^{\frac{3(d-1)}{2}}$ | $\lceil\log_2 d\rceil+1$ | $\mathcal{C}^2$, non-linear | **This paper** |

Table 1: Approximation results for Sobolev $W^{r,p}$ and Korobov $X^{2,\infty}$ functions by shallow and deep networks. Number of neurons and training parameters are given in $\mathcal{O}$ notation.

## 2 PRELIMINARIES

In this work, we consider feed-forward neural networks, using a linear output neuron and a non-linear activation function $\sigma : \mathbb{R} \to \mathbb{R}$ for the other neurons, such as the popular rectified unit (ReLU) $\sigma(x) = \max(x, 0)$, the sigmoid $\sigma(x) = (1 + e^{-x})^{-1}$ or the Heaviside function $\sigma(x) = \mathbf{1}_{\{x \geq 0\}}$. Let $d \geq 1$ be the dimension of the input. We define a 1-hidden-layer network with $N$ neurons as $\boldsymbol{x} \mapsto \sum_{k=1}^{N} u_k \sigma(\boldsymbol{w_k}^\top \boldsymbol{x} + b_k)$, where $\boldsymbol{w_k} \in \mathbb{R}^d$, $b_k \in \mathbb{R}$ for $i = 1, \ldots, N$, are parameters. A neural network with several hidden layers is obtained by feeding the outputs of a given layer as inputs to the next layer. We study the expressive power of neural networks, i.e., the ability to approximate a target function $f : \mathbb{R}^d \to \mathbb{R}$ with as few neurons as possible, on the unit hyper-cube $\Omega := [0, 1]^d$. Another relevant metric is the number of parameters that need to be trained to approximate the function, i.e., the number of parameters of the approximating network ($u_k$, $\boldsymbol{w}_k$ and $b_k$) depending on the function to approximate. We will adopt $L^\infty$ norm as a measure of approximation error.

We now define some notations necessary to introduce our function spaces of interest. For an integer $r$, we denote $\mathcal{C}^r$ the space of one dimensional functions differentiable $r$ times and with continuous derivatives. In our analysis, we consider functions $f$ with bounded mixed derivatives. For a multi-index $\boldsymbol{\alpha} \in \mathbb{N}^d$, the derivative of order $\boldsymbol{\alpha}$ is $D^{\boldsymbol{\alpha}} f := \frac{\partial^{|\boldsymbol{\alpha}|_1} f}{\partial x_1^{\alpha_1} \ldots \partial x_d^{\alpha_d}}$, where $|\boldsymbol{\alpha}|_1 = \sum_{i=1}^{d} |\alpha_i|$.

Two common function spaces in a compact $\Omega \subset \mathbb{R}^d$ are the Sobolev spaces $W^{r,p}(\Omega)$ of functions having weak partial derivatives up to order $r$ in $L^p(\Omega)$ and the Korobov spaces $X^{r,p}(\Omega)$ of functions vanishing at the boundary and having weak mixed second derivatives up to order $r$ in $L^p(\Omega)$.

$$W^{r,p}(\Omega) = \{f \in L^p(\Omega) \ : \ D^{\boldsymbol{\alpha}} f \in L^p(\Omega), |\boldsymbol{\alpha}|_1 \leq r\},$$
$$X^{r,p}(\Omega) = \{f \in L^p(\Omega) \ : \ f|_{\partial\Omega} = 0, D^{\boldsymbol{\alpha}} f \in L^p(\Omega), |\boldsymbol{\alpha}|_\infty \leq r\}.$$

where $\partial\Omega$ denotes the boundary of $\Omega$, $|\boldsymbol{\alpha}|_1 = \sum_{i=1}^{d} |\alpha_i|$ and $|\boldsymbol{\alpha}|_\infty = \sup_{i=1,\ldots,d} |\alpha_i|$ are respectively the $L^1$ and infinity norm. Note that Korobov spaces $X^{r,p}(\Omega)$ are subsets of Sobolev spaces $W^{r,p}(\Omega)$. For $p = \infty$, the usual norms on these spaces are given by

$$|f|_{W^{r,p}(\Omega)} := \max_{|\boldsymbol{\alpha}|_1 \leq r} \|D^{\boldsymbol{\alpha}} f\|_\infty, \quad |f|_{X^{r,p}(\Omega)} := \max_{|\boldsymbol{\alpha}|_\infty \leq r} \|D^{\boldsymbol{\alpha}} f\|_\infty,$$

For simplicity, we will write $|\cdot|_{2,\infty}$ for $|\cdot|_{X^{2,\infty}}$. We focus our analysis on approximating functions on the Korobov space $X^{2,\infty}(\Omega)$ for which the curse of dimensionality is drastically lessened and we show that neural networks are near-optimal. Intuitively, a key difference compared to the Sobolev space is that Korobov functions do not have high frequency oscillations in *all* directions at a time. Such functions may require an exponential number of neurons Telgarsky (2016) and are one of the main difficulties for Sobolev space approximation, which therefore exhibits the curse of

dimensionality DeVore et al. (1989). On the contrary, the Korobov space prohibits such behaviour by ensuring that functions can be differentiable on all dimensions together. Further discussions and concrete examples are given in Appendix A.

## 2.1 THE CURSE OF DIMENSIONALITY

We adopt the point of view of asymptotic results in $\epsilon$ (or equivalently, in the number of neurons), which is a well-established setting in the neural networks representation power literature (Mhaskar, 1996; Bungartz & Griebel, 2004; Yarotsky, 2017; Montanelli & Du, 2019) and numerical analysis literature (Novak, 2006). In the rest of the paper, we use $\mathcal{O}$ notation which hide constants in $d$. For each result, full dependencies in $d$ are provided in appendix. Previous efforts to quantify the number of neurons needed to approximate large general class of functions, showed that neural networks and most classical functional approximation schemes exhibit the curse of dimensionality. For example, for Sobolev functions, Mhaskar proved the following approximation bound.

**Theorem 2.1** (Mhaskar (1996)). *Let $p, r \geq 1$, and $\sigma : \mathbb{R} \to \mathbb{R}$ be an infinitely differentiable activation function, non-polynomial on any interval of $\mathbb{R}$. Let $\epsilon > 0$ sufficiently small. For any $f \in W^{r,p}$, there exists a shallow neural network with one hidden layer, activation function $\sigma$, and $\mathcal{O}(\epsilon^{-\frac{d}{r}})$ neurons approximating $f$ within $\epsilon$ for the infinity norm.*

Therefore, the approximation of Sobolev functions by neural networks suffers from the curse of dimensionality since the number of neurons needed grows exponentially with the input space dimension $d$. This curse is not due to poor performance of neural networks but rather to the choice of the Sobolev space. DeVore et al. (1989) proved that any learning algorithm with continuous parameters needs at least $\Theta(\epsilon^{-\frac{d}{r}})$ parameters to approximate the Sobolev space $W^{r,p}$. This shows that the class of Sobolev functions suffers inherently from the curse of dimensionality and no continuous function approximator can overcome it. We detail this notion later in Section 5.

The natural question is whether there exists a reasonable and sufficiently large class of functions for which there is no inherent curse of dimensionality. Instead of the Sobolev space, we aim to add more regularity to overcome the curse of dimensionality while preserving a reasonably large space. The Korobov space $X^{2,\infty}(\Omega)$—functions with bounded mixed derivatives—is a natural candidate: it is known in the numerical analysis community as a reasonably large space where numerical approximation methods can lessen the curse of dimensionality (Bungartz & Griebel, 2004). Korobov functions have been introduced for solving partial differential equations (Korobov, 1959; Smolyak, 1963), and have then been used extensively for high-dimensional function approximation (Zenger & Hackbusch, 1991; Bungartz & Griebel, 2004). This space of functions is included in the Sobolev space, but still reasonably large as the regularity condition concerns only second order derivatives. Two questions are of interest. First, how many neurons and training parameters a neural network needs to approximate any Korobov function within $\epsilon$ in the $L^\infty$ norm? Second, how do neural networks perform compared to the optimal theoretical rates for Korobov spaces?

## 2.2 SPARSE GRIDS AND HIERARCHICAL BASIS

In this subsection, we introduce sparse grids which will be key in our neural networks constructions. These were introduced by Zenger (1991) and extensively used for high-dimensional function approximation. We refer to Bungartz & Griebel (2004) for a thorough review of the topic.

The goal is to define discrete approximation spaces with basis functions. Instead of a classical uniform grid partition of the hyper-cube $[0, 1]^d$ involving $n^d$ components, where $n$ is the number of partitions in each coordinate, the sparse grid approach uses a smarter partitioning of the cube preserving the approximation accuracy while drastically reducing the number of components of the grid. The construction involves a 1-dimensional mother function $\phi$ which is used to generate all the functions of the basis. For example, a simple choice for the building block $\phi$ is the standard hat function $\phi(x) := (1 - |x|)_+$. The hat function is not the only possible choice. In the latter proofs we will specify which mother function is used, in our case either the interpolates of Deslauriers & Dubuc (1989) (which we define rigorously later in our proofs) or the hat function $\phi$ which can be seen as the Deslaurier-Dubuc interpolet of order 1. These more elaborate mother functions enjoy more smoothness while essentially preserving the approximation power.

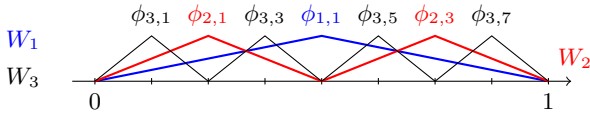

Figure 1: Hierarchical basis from the sparse grid construction using the hat function $(1 - |\cdot|)_+$.

Assume the mother function has support in $[-k, k]$. For $j = 1, \ldots, d$, it can be used to generate a set of local functions $\phi_{l_j, i_j} : [0, 1] \longrightarrow \mathbb{R}$ for all $l_j \geq 1$ and $1 \leq i_j \leq 2^{l_j} - 1$ with support $\left[\frac{i_j - k}{2^{l_j}}, \frac{i_j + k}{2^{l_j}}\right]$ as follows, $\phi_{l_j, i_j}(x) := \phi(2^{l_j} x - i_j)$, $x \in [0, 1]$. We then define a basis of $d$-dimensional functions by taking the tensor product of these 1-dimensional functions. For all $\boldsymbol{l}, \boldsymbol{i} \in \mathbb{N}^d$ with $\boldsymbol{l} \geq \boldsymbol{1}$ and $\boldsymbol{1} \leq \boldsymbol{i} \leq 2^{\boldsymbol{l}} - \boldsymbol{1}$ where $2^{\boldsymbol{l}}$ denotes $(2^{l_1}, \ldots, 2^{l_d})$, define $\phi_{\boldsymbol{l}, \boldsymbol{i}}(\boldsymbol{x}) := \prod_{j=1}^{d} \phi_{l_j, i_j}(x_j)$, $\boldsymbol{x} \in \mathbb{R}^d$. For a fixed $\boldsymbol{l} \in \mathbb{N}^d$, we will consider the *hierarchical increment space* $W_{\boldsymbol{l}}$ which is the subspace spanned by the functions $\{\phi_{\boldsymbol{l}, \boldsymbol{i}} : \boldsymbol{1} \leq \boldsymbol{i} \leq 2^{\boldsymbol{l}} - \boldsymbol{1}\}$, as illustrated in Figure 1,

$$W_{\boldsymbol{l}} := \operatorname{span}\{\phi_{\boldsymbol{l}, \boldsymbol{i}}, \ \boldsymbol{1} \leq \boldsymbol{i} \leq 2^{\boldsymbol{l}} - \boldsymbol{1}, \ i_j \text{ odd for all } 1 \leq j \leq d\}.$$

Note that in the hierarchical increment $W_{\boldsymbol{l}}$, all basis functions have disjoint support. Also, Korobov functions $X^{2,p}(\Omega)$ can be expressed uniquely in this hierarchical basis. Precisely, there is a unique representation of $u \in X^{2,p}(\Omega)$ as $u(\boldsymbol{x}) = \sum_{\boldsymbol{l}, \boldsymbol{i}} v_{\boldsymbol{l}, \boldsymbol{i}} \phi_{\boldsymbol{l}, \boldsymbol{i}}(\boldsymbol{x})$, where the sum is taken over all multi-indices $\boldsymbol{l} \geq \boldsymbol{1}$ and $\boldsymbol{1} \leq \boldsymbol{i} \leq 2^{\boldsymbol{l}} - \boldsymbol{1}$ where all components of $\boldsymbol{i}$ are odd. In particular, all basis functions are linearly independent. Notice that this sum is infinite, the objective is now to define a finite-dimensional subspace of $X^{2,p}(\Omega)$ that will serve as an approximation space. Sparse grids use a carefully chosen subset of the hierarchical basis functions to construct the approximation space $V_n^{(1)} := \bigoplus_{|\boldsymbol{l}|_1 \leq n+d-1} W_{\boldsymbol{l}}$. When $\phi$ is the hat function, Bungartz and Griebel Bungartz & Griebel (2004) showed that this choice of approximating space leads to a good approximating error.

**Theorem 2.2** (Bungartz & Griebel (2004))**.** *Let $f \in X^{2,\infty}(\Omega)$ and $f_n^{(1)}$ be the projection of $f$ on the subspace $V_n^{(1)}$. We have, $\|f - f_n^{(1)}\|_\infty = O\left(2^{-2n} n^{d-1}\right)$. Furthermore, if $v_{\boldsymbol{l}, \boldsymbol{i}}$ denotes the coefficient of $\phi_{\boldsymbol{l}, \boldsymbol{i}}$ in the decomposition of $f_n^{(1)}$ in $V_n^{(1)}$, then we have the upper bound $|v_{\boldsymbol{l}, \boldsymbol{i}}| \leq 2^{-d} 2^{-2|\boldsymbol{l}|_1} |f|_{2,\infty}$, for all $\boldsymbol{l}, \boldsymbol{i} \in \mathbb{N}^d$ with $|\boldsymbol{l}|_1 \leq n+d-1$, $\boldsymbol{1} \leq \boldsymbol{i} \leq 2^{\boldsymbol{l}} - \boldsymbol{1}$ where $\boldsymbol{i}$ has odd components.*

## 3 THE REPRESENTATION POWER OF SHALLOW NEURAL NETWORKS

It has recently been shown that deep neural networks, with depth scaling with $\epsilon$, lessen the curse of dimensionality on the numbers of neuron needed to approximate the Korobov space Montanelli & Du (2019). However, to the best of our knowledge, the question of whether shallow neural networks with fixed universal depth — independent of $\epsilon$ and $d$ — escape the curse of dimensionality as well for the Korobov space remains open. We settle this question by proving that shallow neural networks also lessen the curse of dimensionality for Korobov space.

**Theorem 3.1.** *Let $\epsilon > 0$. For all $f \in X^{2,\infty}(\Omega)$, there exists a neural network with 2 layers, ReLU activation, $\mathcal{O}(\epsilon^{-1}(\log \frac{1}{\epsilon})^{\frac{3(d-1)}{2}+1})$ neurons, and $\mathcal{O}(\epsilon^{-\frac{1}{2}}(\log \frac{1}{\epsilon})^{\frac{3(d-1)}{2}})$ training parameters that approximates $f$ within $\epsilon$ for the infinity norm.*

In order to prove Theorem 3.1, we construct the approximating neural network explicitly. The first step is to construct a neural network architecture with two layers and $\mathcal{O}(d^{\frac{3}{2}} \epsilon^{-\frac{1}{2}} \log \frac{1}{\epsilon})$ neurons that approximates the product function $p : \boldsymbol{x} \in [0,1]^d \longmapsto \prod_{i=1}^{n} x_i$ within $\epsilon$ for all $\epsilon > 0$.

**Proposition 3.2.** *For all $\epsilon > 0$, there exists a neural network with depth 2, ReLU activation and $\mathcal{O}(d^{\frac{3}{2}} \epsilon^{-\frac{1}{2}} \log \frac{1}{\epsilon})$ neurons, that approximates the product function $p : \boldsymbol{x} \in [0,1]^d \longrightarrow \prod_{i=1}^{d} x_i$ within $\epsilon$ for the infinity norm.*

**Sketch of proof** The proof builds upon the observation that $p(\boldsymbol{x}) = \exp(\sum_{i=1}^{d} \log x_i)$. We construct an approximating 2-layer neural network where the first layer approximates $\log x_i$ for $1 \leq i \leq d$, and the second layer approximates the exponential. We illustrate the construction in

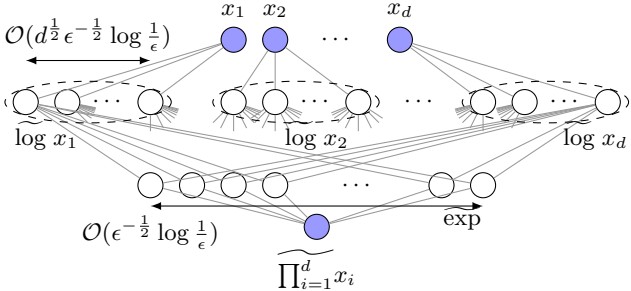

Figure 2: Shallow neural network with ReLU activation implementing the product function $\prod_{i=1}^{d} x_i$ within $\epsilon$ in infinity norm. The network has $\mathcal{O}(d^{\frac{3}{2}}\epsilon^{-\frac{1}{2}}\log\frac{1}{\epsilon})$ neurons on the first layer and $\mathcal{O}(\epsilon^{-\frac{1}{2}}\log\frac{1}{\epsilon})$ neurons on the second layer.

Figure 2. More precisely, fix $\epsilon > 0$. Consider the function $h_\epsilon : x \in [0,1] \mapsto \max(\log x,\ \log \epsilon)$. We approximate $h_\epsilon$ within $\frac{\epsilon}{d}$ with a piece-wise affine function with $\mathcal{O}(d^{\frac{1}{2}}\epsilon^{-\frac{1}{2}}\log\frac{1}{\epsilon})$ pieces, then represent this piece-wise affine function with a single layer neural network $\hat{h}_\epsilon$ with the same number of neurons as the number of pieces (Lemma B.1, Appendix B.1). This 1-layer network then has $\|h_\epsilon - \hat{h}_\epsilon\|_\infty \leq \frac{\epsilon}{d}$. The first layer of our final network is the union of $d$ copies of $\hat{h}_\epsilon$: one for each dimension $i$, approximating $\log x_i$. Similarly, consider the exponential $g : x \in \mathbb{R}_- \mapsto e^x$. We construct a 1-layer neural network $\hat{g}_\epsilon$ with $\mathcal{O}(\epsilon^{-\frac{1}{2}}\log\frac{1}{\epsilon})$ neurons with $\|g - \hat{g}_\epsilon\|_\infty \leq \epsilon$. This will serve as second layer. Formally, the constructed network $\hat{p}_\epsilon$ is $\hat{p}_\epsilon = \hat{g}_\epsilon\left(\sum_{i=1}^{d} \hat{h}_\epsilon(x_i)\right)$. This 2-layer neural network has $\mathcal{O}(d^{\frac{3}{2}}\epsilon^{-\frac{1}{2}}\log\frac{1}{\epsilon})$ neurons and verifies $\|\hat{p}_\epsilon - p\|_\infty \leq \epsilon$.

We use this result to prove Theorem 3.1 and show that we can approximate any Korobov function $f \in X^{2,\infty}(\Omega)$ within $\epsilon$ with a 2-layer neural network of $\mathcal{O}(\epsilon^{-\frac{1}{2}}(\log\frac{1}{\epsilon})^{\frac{3(d-1)}{2}})$ neurons. Consider the sparse grid construction of the approximating space $V_n^{(1)}$ using the standard hat function as mother function to create the hierarchical basis $W_{\boldsymbol{l}}$ (introduced in Section 2.2). The key idea is to construct a shallow neural network approximating the sparse grid approximation and then use the result of Theorem 2.2 to derive the approximation error. Let $f_n^{(1)}$ be the projection of $f$ on the subspace $V_n^{(1)}$ defined in Section 2.2. $f_n^{(1)}$ can be written as $f_n^{(1)}(\boldsymbol{x}) = \sum_{(\boldsymbol{l},\boldsymbol{i})\in U_n^{(1)}} v_{\boldsymbol{l},\boldsymbol{i}}\phi_{\boldsymbol{l},\boldsymbol{i}}(\boldsymbol{x})$, where $U_n^{(1)}$ contains the indices $(\boldsymbol{l},\boldsymbol{i})$ of basis functions present in $V_n^{(1)}$. We can use Theorem 2.2 and choose $n$ carefully such that $f_n^{(1)}$ approximates $f$ within $\epsilon$ for $L^\infty$ norm. The goal is now to approximate $f_n^{(1)}$ with a shallow neural network. Note that the basis functions can be written as a product of univariate functions $\phi_{\boldsymbol{l},\boldsymbol{i}} = \prod_{j=1}^{d} \phi_{l_j,i_j}$. We can therefore use a similar structure to the product approximation of Proposition 3.2 to approximate the basis functions. Specifically, the first layer approximate the $d(2^n - 1) = \mathcal{O}(\epsilon^{-\frac{1}{2}}(\log\frac{1}{\epsilon})^{\frac{d-1}{2}})$ terms $\log\phi_{l_j,i_j}$ necessary to construct the basis functions of $V_n^{(1)}$ and a second layer to approximate the exponential in order to obtain approximations of the $\mathcal{O}(2^n n^{d-1}) = \mathcal{O}(\epsilon^{-\frac{1}{2}}(\log\frac{1}{\epsilon})^{\frac{3(d-1)}{2}})$ basis functions of $V_n^{(1)}$. We provide a detailed figure illustrating the construction, Figure 5 in Appendix B.3.

The shallow network that we constructed in Theorem 3.1 uses the ReLU activation function. We extend this result to a larger class of activation functions which include commonly used ones.

**Definition 3.3.** *A* sigmoid-like *activation function* $\sigma : \mathbb{R} \to \mathbb{R}$ *is a non-decreasing function having finite limits in* $\pm\infty$. *A* ReLU-like *activation function* $\sigma : \mathbb{R} \to \mathbb{R}$ *is a function having a horizontal asymptote in* $-\infty$ *i.e.* $\sigma$ *is bounded in* $\mathbb{R}_-$, *and an affine (non-horizontal) asymptote in* $+\infty$, *i.e. there exists* $b > 0$ *such that* $\sigma(x) - bx$ *is bounded in* $\mathbb{R}_+$.

Most common activation functions fall into these classes. Examples of sigmoid-like activations include Heaviside, logistic, tanh, arctan and softsign activations, while ReLU-like activations include ReLU, ISRLU, ELU and soft-plus activations. We extend Theorem 3.1 to all these activations.

**Theorem 3.4.** *For any approximation tolerance $\epsilon > 0$, and for any $f \in X^{2,\infty}(\Omega)$ there exists a neural network with depth 2 and $\mathcal{O}(\epsilon^{-\frac{1}{2}}(\log\frac{1}{\epsilon})^{\frac{3(d-1)}{2}})$ training parameters that approximates $f$ within $\epsilon$ for the infinity norm, with $O\left(\epsilon^{-1}\log(\frac{1}{\epsilon})^{\frac{3(d-1)}{2}+1}\right)$ (resp. $\mathcal{O}(\epsilon^{-\frac{3}{2}}\log(\frac{1}{\epsilon})^{\frac{3(d-1)}{2}})$) neurons for a ReLU-like (resp. sigmoid-like) activation.*

We note that these results can further be extended to more general Korobov spaces $X^{r,p}$. Indeed, the main dependence of our neural network architectures in the parameters $r$ and $p$ arise from sparse grid approximation. Bungartz & Griebel (2004) show that results similar to Theorem 2.2 can be extended to various values of $r$, $p$ and different error norms with a similar sparse grid construction. For instance, we can use these results combined with our proposed architecture to show that the Korobov space $X^{r,\infty}$ can be approximated in infinite norm by neural networks with $O(\epsilon^{-\frac{1}{r}}(\log\frac{1}{\epsilon})^{\frac{r+1}{r}(d-1)})$ training parameters and same number of neurons up to a polynomial factor in $\epsilon$.

# 4 THE REPRESENTATION POWER OF DEEP NEURAL NETWORKS

Montanelli & Du (2019) used the sparse grid approach to construct deep neural networks with ReLU activation, approximating Korobov functions with $\mathcal{O}(\epsilon^{-\frac{1}{2}}(\log\frac{1}{\epsilon})^{\frac{3(d-1)}{2}+1})$ neurons, and depth $\mathcal{O}(\log\frac{1}{\epsilon})$ for the $L^\infty$ norm. We improve this bound for deep neural networks with $\mathcal{C}^2$ non-linear activation functions. We prove that we only need $\mathcal{O}(\epsilon^{-\frac{1}{2}}(\log\frac{1}{\epsilon})^{\frac{3(d-1)}{2}})$ neurons and fixed depth, independent of $\epsilon$, to approximate the unit ball of the Korobov space within $\epsilon$ in the $L^\infty$ norm.

**Theorem 4.1.** *Let $\sigma \in \mathcal{C}^2$ be a non-linear activation function. Let $\epsilon > 0$. For any function $f \in X^{2,\infty}(\Omega)$, there exists a neural network of depth $\lceil \log_2 d \rceil + 1$, with ReLU activation on the first layer and activation function $\sigma$ for the next layers, $\mathcal{O}(\epsilon^{-\frac{1}{2}}(\log\frac{1}{\epsilon})^{\frac{3(d-1)}{2}})$ neurons, and $\mathcal{O}(\epsilon^{-\frac{1}{2}}(\log\frac{1}{\epsilon})^{\frac{3(d-1)}{2}})$ training parameters approximating $f$ within $\epsilon$ for the infinity norm.*

Compared to the bound of shallow networks in Theorem 3.1, the number of neurons for deep networks is lower by a factor $\mathcal{O}(\sqrt{\epsilon})$, while the number of training parameters is the same. Hence, deep neural networks are more efficient than shallow neural network in the sense that shallow networks need more "inactive" neurons to reach the same approximation power, but have the same number of parameters. This gap in the number of "inactive" neurons can be consequent in practice, as we may not know exactly which neurons to train and which neurons to fix.

This new bound on the number of parameters and neurons matches the approximation power of sparse grids. In fact, sparse grids use $\Theta(\epsilon^{-\frac{1}{2}}(\log\frac{1}{\epsilon})^{\frac{3(d-1)}{2}})$ parameters (weights of basis functions) to approximate Korobov functions within $\epsilon$. Our construction in Theorem 4.1 shows that deep neural networks with fixed depth in $\epsilon$ can fully encode sparse grids approximators. Neural networks are therefore more powerful function approximators. In particular, any sparse grid approximation using $\mathcal{O}(N(\epsilon))$ parameters, can be represented exactly by a neural network using $\mathcal{O}(N(\epsilon))$ neurons.

The deep approximating network (see Figure 3) has a very similar structure to our construction of an approximating shallow network of Theorem 3.1. The main difference lies in the approximation of the product function. Instead of using a 2-layer neural network, we now use a deep network. The following result shows that deep neural networks can represent exactly the product function.

**Proposition 4.2** (Lin et al. (2017), Appendix A). *Let $\sigma$ be $\mathcal{C}^2$ non linear activation function. For any approximation error $\epsilon > 0$, there exists a neural network with $\lceil \log_2 d \rceil$ hidden layers and activation $\sigma$, using at most $8d$ neurons arranged in a binary tree network that approximates the product function $\prod_{i=1}^{d} x_i$ on $[0,1]^d$ within $\epsilon$ for the infinity norm.*

An important remark is that the structure of the constructed neural network is independent of $\epsilon$. In particular, the depth and number of neurons is independent of the approximation precision $\epsilon$, which we refer to as *exact* approximation. It is known that an exponential number of neurons is needed in order to *exactly* approximate the product function with a 1-layer neural network Lin et al. (2017), however, the question of whether one could approximate the product with a shallow network and a polynomial number of neurons, remained open. In Proposition 3.2, we answer positively to this question by constructing an $\epsilon$-approximating neural network of depth 2 with ReLU activation and $\mathcal{O}(d^{\frac{3}{2}}\epsilon^{-\frac{1}{2}}\log\frac{1}{\epsilon})$ neurons. Using the same ideas as in Theorem 3.4, we can generalize this result to

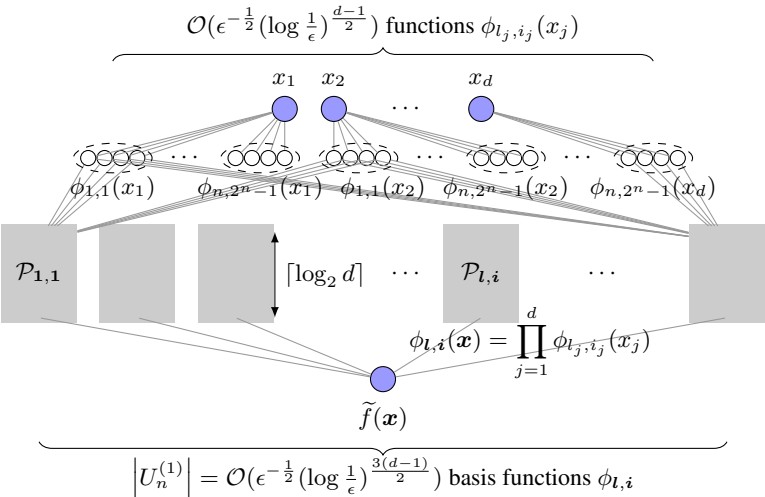

Figure 3: Deep neural network approximating a Korobov function $f \in X^{2,\infty}(\Omega)$ within $\epsilon$. The complete network has $\mathcal{O}(\epsilon^{-\frac{1}{2}}(\log \frac{1}{\epsilon})^{\frac{3(d-1)}{2}})$ neurons and depth $\lceil \log_2 d \rceil + 1$.

obtain an $\epsilon$-approximating neural network of depth 2 with $\mathcal{O}(d^{\frac{3}{2}}\epsilon^{-\frac{1}{2}} \log \frac{1}{\epsilon})$ neurons for a ReLU-like activation, or $\mathcal{O}(d^2\epsilon^{-1} \log \frac{1}{\epsilon})$ neurons for a sigmoid-like activation.

## 5 NEURAL NETWORKS ARE NEAR-OPTIMAL FUNCTION APPROXIMATORS

In the previous sections, we proved upper bounds on the number of neurons and training parameters needed by deep and shallow neural networks to approximate the Korobov space $X^{2,\infty}(\Omega)$. We now investigate how good is the performance of neural networks as function approximators. We prove a lower bound on the number of parameters needed by any continuous function approximator to approximate the Korobov space. In particular, neural networks, deep and shallow, will nearly match this lower bound, making them near-optimal function approximators. Let us first formalize the notion of continuous function approximators, following the framework of DeVore et al. (1989).

For any Banach space $\mathcal{X}$—e.g., a function space—and a subset $K \subset \mathcal{X}$ of elements to approximate, we define a continuous function approximator with $N$ parameters as a continuous parametrization $a : K \to \mathbb{R}^N$ together with a reconstruction scheme which is a $N$-dimensional manifold $\mathcal{M}_N : \mathbb{R}^N \to \mathcal{X}$. For any element $f \in K$, the approximation given is $\mathcal{M}_N(a(f))$: the parametrization $a$ is derived continuously from the function $f$ and then given as input to the reconstruction manifold that outputs an approximation function in $\mathcal{X}$. The error of this function approximator is defined as $E_{N,a,\mathcal{M}_N}(K)_{\mathcal{X}} := \sup_{f \in K} |f - \mathcal{M}_N(a(f))|_{\mathcal{X}}$. The best function approximator for space $K$ minimizes this error. The minimal error for space $K$ is given by

$$E_N(K)_{\mathcal{X}} = \min_{a, \mathcal{M}_N} E_{N,a,\mathcal{M}_N}(K)_{\mathcal{X}}.$$

In other terms, a continuous function approximator with $N$ parameters cannot hope to approximate $K$ better than within $E_N(K)_{\mathcal{X}}$. A class of function approximators is a set of function approximators with a given structure. For example, neural networks with continuous parametrizations are a class of function approximators where the number of parameters is the number of training parameters. We say that a class of function approximators is optimal for the space of functions $K$ if it matches this minimal error asymptotically in $N$, within a constant multiplicative factor. In other words, the number of parameters needed by the class to approximate functions in $K$ within $\epsilon$ matches asymptotically, within a constant, the least number of parameters $N$ needed to satisfy $E_N(K)_{\mathcal{X}} \le \epsilon$. The norm considered in the approximation of the functions of $K$ is the norm associated to the space $\mathcal{X}$. DeVore et al. (1989) showed that this minimum error $E_N(K)_{\mathcal{X}}$ is lower bounded by the Bernstein width of the subset $K \subset \mathcal{X}$, defined as

$$b_N(K)_{\mathcal{X}} := \sup_{X_{N+1}} \sup\{\rho : \ \rho U(X_{N+1}) \subset K\},$$

where the outer sup is taken over all $N + 1$ dimensional linear sub-spaces of $\mathcal{X}$, and $U(Y)$ denotes the unit ball of $Y$ for any linear subspace $Y$ of $\mathcal{X}$.

**Theorem 5.1** (DeVore et al. (1989))**.** *Let $\mathcal{X}$ a Banach space and $K \subset \mathcal{X}$. $E_N(K)_{\mathcal{X}} \geq b_N(K)_{\mathcal{X}}$.*

We prove a lower bound on the least number of parameters any class of continuous function approximators needs to approximate functions of the Korobov space.

**Theorem 5.2.** *Take $\mathcal{X} = L^{\infty}(\Omega)$ and $K = \{f \in X^{2,\infty}(\Omega) : |f|_{X^{2,\infty}(\Omega)} \leq 1\}$ the unit ball of the Korobov space. Then, there exists $c > 0$ with $E_N(K)_{\mathcal{X}} \geq \frac{c}{N^2}(\log N)^{d-1}$. Equivalently, for $\epsilon > 0$, a continuous function approximator approximating $K$ within $\epsilon$ in $L^{\infty}$ norm uses at least $\Theta(\epsilon^{-\frac{1}{2}}(\log \frac{1}{\epsilon})^{\frac{d-1}{2}})$ parameters.*

**Sketch of proof** We seek an appropriate subspace $X_{N+1}$ in order to get lower bound the Bernstein width $b_N(K)_{\mathcal{X}}$, which in turn provides a lower bound on the approximation error (Theorem 5.1). To do so, we use the Deslaurier-Dubuc interpolet of degree 2, $\phi^{(2)}$ (see Figure 6) which is $\mathcal{C}^2$. Using the sparse grids approach, we construct a hierarchical basis in $X^{2,\infty}(\Omega)$ using $\phi^{(2)}$ as mother function and define $X_{N+1}$ as the approximation space $V_n^{(1)}$. Here $n$ is chosen such that the dimension of $V_n^{(1)}$ is roughly $N + 1$. The goal is to estimate $\sup\{\rho : \rho U(X_{N+1}) \subset K\}$, which will lead to a bound on $b_N(K)_{\mathcal{X}}$. To do so, we upper bound the Korobov norm by the $L^{\infty}$ norm for elements of $X_{N+1}$. For any function $u \in X_{N+1}$ we can write $u = \sum_{\boldsymbol{l},\boldsymbol{i}} v_{\boldsymbol{l},\boldsymbol{i}} \cdot \phi_{\boldsymbol{l},\boldsymbol{i}}$. Using a stencil representation of the coefficients $v_{\boldsymbol{l},\boldsymbol{i}}$, we are able to obtain an upper bound $|u|_{X^{2,\infty}} \leq \Gamma_d \|u\|_{\infty}$ where $\Gamma_d = \mathcal{O}(2^{2n}n^{d-1})$. Then, $b_N(K)_{\mathcal{X}} \geq 1/\Gamma_d$ which yields the desired bound.

This lower bound matches within a logarithmic factor the upper bound on the number of training parameters needed by deep and shallow neural networks to approximate the Korobov space within $\epsilon$: $\mathcal{O}(\epsilon^{-\frac{1}{2}}(\log \frac{1}{\epsilon})^{\frac{3(d-1)}{2}})$ (Theorem 3.1 and Theorem 4.1). It exhibits the same exponential dependence in $d$ with base $\log \frac{1}{\epsilon}$ and the same main dependence on $\epsilon$ of $\epsilon^{-\frac{1}{2}}$. Note that the upper and lower bound can be rewritten as $\mathcal{O}(\epsilon^{-1/2-\delta})$ for all $\delta > 0$. Moreover, our constructions in Theorem 3.1 and Theorem 4.1 are continuous, which comes directly from the continuity of the sparse grid parameters (see bound on $v_{\boldsymbol{l},\boldsymbol{i}}$ in Theorem 2.2). Our bounds prove therefore that deep and shallow neural networks are near optimal classes of function approximators for the Korobov space. Interestingly, the subspace $X_{N+1}$ our proof uses to show the lower bound is essentially the same as the subspace we use to approximate Koborov functions in our proof of the upper bounds (Theorem 3.1 and 4.1). The difference is in the choice of the interpolate $\phi$ to construct the basis functions, degree 2 for the former (which provides needed regularity for the proof), and 1 for the later.

## 6 CONCLUSION AND DISCUSSION

We proved new upper and lower bounds on the number of neurons and training parameters needed by shallow and deep neural networks to approximate Korobov functions. Our work shows that shallow and deep networks not only lessen the curse of dimensionality but are also near-optimal.

Our work suggests several extensions. First, it would be very interesting to see if our proposed theoretical near-optimal architectures have powerful empirical performance. While commonly used structures (e.g. Convolution Neural Networks, or Recurrent Neural Networks) are motivated by properties of the data such as symmetries, our structures are motivated by theoretical insights on how to optimally approximate a large class of functions with a given number of neurons and parameters. Second, our upper bounds (Theorem 3.1 and 4.1) nearly match our lower bounds (Theorem 5.2) on the least number of training parameters needed to approximate the Korobov space. We wonder if it is possible to close the gap between these bounds and hence prove neural network's optimality, e.g., one could prove that sparse grids are optimal function approximators by improving our lower bound to match sparse grid number of parameters $\mathcal{O}(\epsilon^{-\frac{1}{2}}(\log \frac{1}{\epsilon})^{\frac{3(d-1)}{2}})$. Finally, we showed the near-optimality of neural networks among the set of continuous function approximators. It would be interesting to explore lower bounds (analog to Theorem 5.2) when considering larger sets of function approximators, e.g., discontinuous function approximators. Could some discontinuous neural network construction break the curse of dimensionality for the Sobolev space? The question is then whether neural networks are still near-optimal in these larger sets of function approximators.

## ACKNOWLEDGMENTS

The authors are grateful to Tomaso Poggio and the MIT 6.520 course teaching staff for several discussions, remarks and comments that were useful to this work.

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

## APPENDIX

## A ON KOROBOV FUNCTIONS

In this section, we further discuss Korobov functions $X^{2,p}(\Omega)$. Korobov functions enjoy more smoothness than Sobolev functions: smoothness for $X^{2,p}(\Omega)$ is measured in terms of *mixed derivatives* of order two. Korobov functions $X^{2,p}(\Omega)$ can be differentiated twice in *each* coordinates simultaneously, while Sobolev $W^{2,p}(\Omega)$ functions can only be differentiated twice in total. For example, in two dimensions, for a function $f$ to be Korobov it is required to have

$$\frac{\partial f}{\partial x_1}, \frac{\partial f}{\partial x_2}, \frac{\partial^2 f}{\partial^2 x_1}, \frac{\partial^2 f}{\partial^2 x_2}, \frac{\partial^2 f}{\partial x_1 \partial x_2}, \frac{\partial^3 f}{\partial^2 x_1 \partial x_2}, \frac{\partial^3 f}{\partial x_1 \partial^2 x_2}, \frac{\partial^4 f}{\partial^2 x_1 \partial^2 x_2} \in L^p(\Omega),$$

while for $f$ to be Sobolev it requires only

$$\frac{\partial f}{\partial x_1}, \frac{\partial f}{\partial x_2}, \frac{\partial^2 f}{\partial^2 x_1}, \frac{\partial^2 f}{\partial^2 x_2}, \frac{\partial^2 f}{\partial x_1 \partial x_2} \in L^p(\Omega).$$

The former can be seen from $|\boldsymbol{\alpha}|_\infty \leq 2$ and the latter from $|\boldsymbol{\alpha}|_1 \leq 2$ in the definition of $X^{r,p}(\Omega)$ and $W^{r,p}(\Omega)$.

We now provide intuition on why Korobov functions are easier to approximate. One of the key difficulties in approximating Sobolev functions are possible high frequency oscillations which may require an exponential number of neurons (Telgarsky, 2016). For instance, consider functions which have similar structure to $W_{(n...n)}$ (defined in Subsection 2.2): for any smooth basis function $\phi$ with support on the unit cube (see Figure 6 for example), consider the linear function space formed by linear combinations of dilated function $\phi$ with support on each cube $d$-dimensional grid of step $2^{-n}$. This corresponds exactly to the construction of $W_{(n...n)}$ which uses the product of hat function on each dimension as basis function $\phi$. This function space can have strong oscillations in *all* directions at a time. The Korobov space prohibits such behavior by ensuring that functions can be differentiated twice on each dimension, simultaneously. As a result, functions cannot oscillate in all directions at a time without having large Korobov norm. We end this paragraph by comparing the Korobov space to the space of bandlimited functions which was shown to avoid the curse of dimensionality (Montanelli et al., 2019). These are functions for which the support frequency components is restricted to a fixed compact. Intuitively, approximating these functions can be achieved because the set of frequencies is truncated to a compact, which then allows to sample frequencies and obtain approximation guarantees. Instead of imposing a hard constraint of cutting high frequencies, the Korobov space asks for smoothness conditions which do not prohibit high frequencies but rather impose a budget for high frequency oscillations. We precise this idea in the next example.

A concrete example of Korobov functions is given by an analogous of the function space $V_n^{(1)}$ which we used as approximation space in the proof of our results (see Section 2.2). Similarly to the previous paragraph, one should use a smooth basis function to ensure differentiability. Recall that $V_n^{(1)}$ is defined as

$$V_n^{(1)} := \bigoplus_{|\boldsymbol{l}|_1 \leq n+d-1} W_{\boldsymbol{l}}.$$

Intuitively, this approximation space introduces a "budget" of oscillations for *all* dimensions through the constraint $\sum_{i=1}^{d} l_i \leq n + d - 1$. As a result, dilations of the basis function can only occur in a restricted set of directions at a time, which ensures that the Korobov norm stays bounded.

# B   PROOFS OF SECTION 3

## B.1   APPROXIMATING THE PRODUCT FUNCTION

In this subsection, we construct a neural network architecture with two layers and $\mathcal{O}(d^{\frac{3}{2}}\epsilon^{-\frac{1}{2}}\log\frac{1}{\epsilon})$ neurons that approximates the product function $p : \boldsymbol{x} \in [0,1]^d \longmapsto \prod_{i=1}^{n} x_i$ within $\epsilon$ for all $\epsilon > 0$, which proves Proposition 3.2. We first prove a simple lemma to represent univariate piece-wise affine functions by shallow neural networks.

**Lemma B.1.** *Any one dimensional continuous piece-wise affine function with $m$ pieces is representable exactly by a shallow neural network with ReLU activation, with $m$ neurons on a single layer.*

*Proof.* This is a simple consequence from Proposition 1 in Yarotsky (2017). We recall the proof for completeness. Let $x_1 \leq \cdots \leq x_{m-1}$ be the subdivision of the piece-wise affine function $f$. We use a neural network of the form

$$g(x) := f(x_1) + \sum_{k=1}^{m-1} w_k(x - x_k)_+ - w_0(x_1 - x)_+,$$

where $w_0$ is the slope of $f$ on the piece $\leq x_1$, $w_1$ is the slope of $f$ on the piece $[x_1, x_2]$,

$$w_k = \frac{f(x_{k+1}) - f(x_1) - \sum_{i=1}^{k-1} w_i(x_{k+1} - x_i)}{x_{k+1} - x_k},$$

for $k = 1, \cdots, m-2$, and $w_{m-1} = \tilde{w} - \sum_{k=1}^{m-2} w_k$ where $\tilde{w}$ is the slope of $f$ on the piece $\geq x_{m-1}$. Notice that $f$ and $g$ coincide on all $x_k$ for $1 \leq k \leq m-1$. Furthermore, $g$ has same slope as $f$ on each pieces, therefore, $g = f$. $\qquad\square$

We can approximate univariate right continuous functions by piece-wise affine functions, and then use Lemma B.1 to represent them by shallow neural networks. The following lemma shows that $\mathcal{O}(\epsilon^{-1})$ neurons are sufficient to represent an increasing right-continuous function with a shallow neural network.

**Lemma B.2.** *Let $f : I \longrightarrow [c, d]$ be a right-continuous increasing function where $I$ is an interval, and let $\epsilon > 0$. There exists a shallow neural network with ReLU activation, with $\left\lceil \frac{d-c}{\epsilon} \right\rceil$ neurons on a single layer, that approximates $f$ within $\epsilon$ for the infinity norm.*

*Proof.* Let $m = \left\lfloor \frac{d-c}{\epsilon} \right\rfloor$ Define a subdivision of the image interval $c \leq y_1 \leq \ldots \leq y_m \leq d$ where $y_k = c + k\epsilon$ for $k = 1, \ldots, m$. Note that this subdivision contains exactly $\left\lceil \frac{d-c}{\epsilon} \right\rceil$ pieces. Now define a subdivision of $I$, $x_1 \leq x_2 \leq \ldots \leq x_m$ by

$$x_k := \sup\{x \in I, f(x) \leq y_k\},$$

for $k = 1, \ldots, m$. This subdivision stills has $\left\lceil \frac{d-c}{\epsilon} \right\rceil$ pieces. We now construct our approximation function $\hat{f}$ on $I$ as the continuous piece-wise affine function on the subdivision $x_1 \leq \ldots \leq x_m$ such that $\hat{f}(x_k) = y_k$ for all $1 \leq k \leq m$ and $\hat{f}$ is constant before $x_1$ and after $x_m$ (see Figure 4). Let $x \in I$.

- If $x \leq x_1$, because $f$ is increasing and right-continuous, $c \leq f(x) \leq f(x_1) \leq y_1 = c + \epsilon$. Therefore $|f(x) - \hat{f}(x)| = |f(x) - (c + \epsilon)| \leq \epsilon$.

- If $x_k < x \leq x_{k+1}$, we have $y_k < f(x) \leq f(x_{k+1}) \leq y_{k+1}$. Further note that $y_k \leq \hat{f}(x) \leq y_{k+1}$. Therefore $|f(x) - \hat{f}(x)| \leq y_{k+1} - y_k = \epsilon$.

- If $x_m < x$, then $y_m < f(x) \leq d$. Again, $|f(x) - \hat{f}(x)| = |f(x) - y_m| \leq d - y_m \leq \epsilon$.

Therefore $\|f - \hat{f}\|_\infty \leq \epsilon$. We can now use Lemma B.1 to end the proof. $\qquad\square$

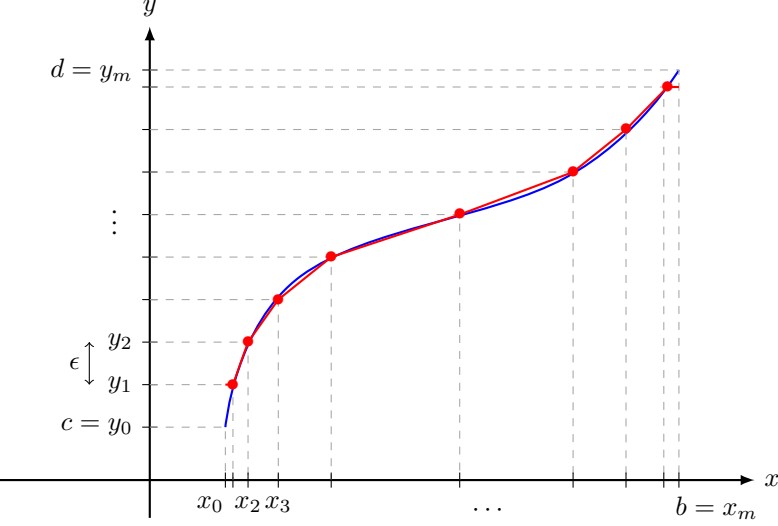

Figure 4: Approximation of a right-continuous increasing function (blue) in an interval $[c, d]$ within $\epsilon$ by a piece-wise linear function (red) with $\left\lfloor \frac{d-c}{\epsilon} \right\rfloor$ pieces. The approximation is constructed using a regular subdivision of the $y$ axis of step $\epsilon$ and constructing a linear approximation in the pre-image of each part of the subdivision.

If the function to approximate has some regularity, the number of neurons needed for approximation can be significantly reduced. In the following lemma, we show that $\mathcal{O}(\epsilon^{-\frac{1}{2}})$ neurons are sufficient to approximate a $\mathcal{C}^2$ univariate function with a shallow neural network.

**Lemma B.3.** *Let $f : [a, b] \longrightarrow [c, d] \in \mathcal{C}^2$, and let $\epsilon > 0$. There exists a shallow neural network with ReLU activation, with $\frac{1}{\sqrt{2\epsilon}} \min(\int \sqrt{|f''|}(1 + \mu(f, \epsilon)), (b - a)\sqrt{\|f''\|_\infty})$ neurons on a single layer, where $\mu(f, \epsilon) \to 1$ as $\epsilon \to 0$, that approximates $f$ within $\epsilon$ for the infinity norm.*

*Proof.* See Appendix B.2.1. □

We will now use the ideas of Lemma B.2 and Lemma B.3 to approximate a truncated $\log$ function, which we will use in the construction of our neural network approximating the product.

**Corollary B.4.** *Let $\epsilon > 0$ sufficiently small and $\delta > 0$. Consider the truncated logarithm function $\log : [\delta, 1] \longrightarrow \mathbb{R}$. There exists a shallow neural network with ReLU activation, with $\epsilon^{-\frac{1}{2}} \log \frac{1}{\delta}$ neurons on a single layer, that approximates $f$ within $\epsilon$ for the infinity norm.*

*Proof.* See Appendix B.2.2. □

We are now ready to construct a neural network approximating the product function and prove Proposition 3.2. The proof builds upon on the observation that $\prod_{i=1}^{d} x_i = \exp(\sum_{i=1}^{d} \log x_i)$. We construct an approximating 2-layer neural network where the first layer computes $\log x_i$ for $1 \le i \le d$, and the second layer computes the exponential. We illustrate the construction of the proof in Figure 2.

**Proof of Proposition 3.2** Fix $\epsilon > 0$. Consider the function $h_\epsilon : x \in [0, 1] \mapsto \max(\log x, \log \epsilon) \in [\log \epsilon, 0]$. Using Corollary B.4, there exists a neural network $\hat{h}_\epsilon : [0, 1] \longrightarrow [\log \epsilon, 0]$ with $1 + \lceil d^{\frac{1}{2}} \epsilon^{-\frac{1}{2}} \log \frac{1}{\epsilon} \rceil$ neurons on a single layer such that $\|h_\epsilon - \hat{h}_\epsilon\|_\infty \le \frac{\epsilon}{d}$. Indeed, one can take the $\epsilon$-approximation of $h_\epsilon : x \in [\epsilon, 1] \mapsto \log x \in [\log \epsilon, 0]$, then extend this function to $[0, \epsilon]$ with a constant equal to $\log \epsilon$. The resulting piece-wise affine function has one additional segment corresponding to one additional neuron in the approximating function. Similarly, consider the exponential $g : x \in \mathbb{R}_- \mapsto e^x \in [0, 1]$. Because $g$ is right-continuous increasing, we can use Lemma B.3 to construct a neural network $\hat{g}_\epsilon : \mathbb{R}_- \longrightarrow [0, 1]$ with $1 + \lceil \frac{1}{\sqrt{2\epsilon}} \log \frac{1}{\epsilon} \rceil$ neurons on a single layer such that $\|g - \hat{g}_\epsilon\|_\infty \le \epsilon$. Indeed, again one can take the $\epsilon$-approximation of $g_\epsilon : x \in [\log \epsilon, 0] \mapsto e^x \in [0, 1]$, then extend this function to $(-\infty, \log \epsilon]$ with a constant equal to $\epsilon$. The corresponding neural network has an additional neuron. We construct our final neural network $\hat{\phi}_\epsilon$ (see Figure 2) as

$$\hat{\phi}_\epsilon = \hat{g}_\epsilon \left( \sum_{i=1}^{d} \hat{h}_\epsilon(x_i) \right).$$

Note that $\hat{\phi}_\epsilon$ can be represented as a 2-layer neural network: the first layer is composed of the union of the $1 + \lceil d^{\frac{1}{2}} \epsilon^{-\frac{1}{2}} \log \frac{1}{\epsilon} \rceil$ neurons composing each of the 1-layer neural networks $\hat{h}_\epsilon^i : \boldsymbol{x} \in [0, 1]^d \mapsto \hat{h}_\epsilon(x_i) \in \mathbb{R}$ for each dimension $i \in \{1, \ldots, d\}$. The second layer is composed of the $1 + \lceil \frac{1}{\sqrt{2\epsilon}} \log \frac{1}{\epsilon} \rceil$ neurons of $\hat{g}_\epsilon$. Hence, the constructed neural network $\hat{\phi}_\epsilon$ has $\mathcal{O}(d^{\frac{3}{2}} \epsilon^{-\frac{1}{2}} \log \frac{1}{\epsilon})$ neurons. Let us now analyze the approximation error. Let $\boldsymbol{x} \in [0, 1]^d$. For the sake of brevity, denote $\hat{y} = \sum_{i=1}^{d} \hat{h}_\epsilon(x_i)$ and $y = \sum_{i=1}^{d} \log(x_i)$. We have,

$$|\hat{\phi}_\epsilon(\boldsymbol{x}) - p(\boldsymbol{x})| \le |\hat{\phi}_\epsilon(\boldsymbol{x}) - \exp(\hat{y})| + |\exp(\hat{y}) - \exp(y)| \le \epsilon + \prod_{i=1}^{d} x_i \cdot |\exp(\hat{y} - y) - 1|,$$

where we used the fact that $|\hat{\phi}_\epsilon(\boldsymbol{x}) - \exp(\hat{y})| = |\hat{g}_\epsilon(\hat{y}) - g(\hat{y})| \le \|\hat{g}_\epsilon - g\|_\infty \le \epsilon$.

First suppose that $\boldsymbol{x} \ge \epsilon$. In this case, for all $i \in \{1, \ldots, d\}$ we have $|\hat{h}_\epsilon(x_i) - \log(x_i)| = |\hat{h}_\epsilon(x_i) - h_\epsilon(x_i)| \le \frac{\epsilon}{d}$. Then, $|\hat{y} - y| \le \epsilon$. Consequently, $|\hat{\phi}_\epsilon(\boldsymbol{x}) - p(\boldsymbol{x})| \le \epsilon + \max(|e^\epsilon - 1|, |e^{-\epsilon} - 1|) \le 3\epsilon$, for $\epsilon > 0$ sufficiently small. Without loss of generality now suppose $x_1 \le \epsilon$. Then $\hat{y} \le h_\epsilon(x_1) \le \log \epsilon$, so by definition of $\hat{g}_\epsilon$, we have $0 \le \hat{\phi}_\epsilon(\boldsymbol{x}) = \hat{g}_\epsilon(\hat{y}) \le \exp(\log \epsilon) = \epsilon$. Also, $0 \le p(x) \le \epsilon$ so finally $|\hat{\phi}_\epsilon(\boldsymbol{x}) - p(\boldsymbol{x})| \le \epsilon$.

**Remark B.5.** *Note that using Lemma B.2 instead of Lemma B.3 to construct approximating shallow networks for* log *and* exp *would yield approximation functions $\hat{h}_\epsilon$ with $\mathcal{O}(\lceil \frac{d}{\epsilon} \log \frac{1}{\epsilon} \rceil)$ neurons and $\hat{g}_\epsilon$ with $\mathcal{O}(\lceil \frac{1}{\epsilon} \rceil)$ neurons. Therefore, the corresponding neural network would approximate the product $p$ with $\mathcal{O}(d^2 \epsilon^{-1} \log \frac{1}{\epsilon})$ neurons.*

## B.2 MISSING PROOFS OF SECTION B.1

### B.2.1 PROOF OF LEMMA B.3

*Proof.* Similarly as the proof of Lemma B.2, the goal is to approximate $f$ by a piece-wise affine function $\hat{f}$ defined on a subdivision $x_0 = a \leq x_1 \leq \ldots \leq x_m \leq x_{m+1} = b$ such that $f$ and $\hat{f}$ coincide on $x_0, \ldots, x_{m+1}$. We first analyse the error induced by a linear approximation of the function on each piece. Let $x \in [u, v]$ for $u, v \in I$. Using the mean value theorem, there exists $\alpha_x \in [u, x]$ such that $f(x) - f(u) = f'(\alpha_x)(x - u)$ and $\beta_x \in [x, v]$ such that $f(v) - f(x) = f'(\beta_x)(v - x)$. Combining these two equalities, we get,

$$f(x) - f(u) - (x - u)\frac{f(v) - f(u)}{v - u} = \frac{(v - x)(f(x) - f(u)) - (x - u)(f(v) - f(x))}{v - u}$$

$$= (x - u)(x - v)\frac{f'(\beta_x) - f'(\alpha_x)}{v - u}$$

$$= (x - u)(v - x)\frac{\int_{\alpha_x}^{\beta_x} f''(t)dt}{v - u}$$

Hence,

$$f(x) = f(u) + (x - u)\frac{f(v) - f(u)}{v - u} + (x - u)(v - x)\frac{\int_{\alpha_x}^{\beta_x} f''(t)dt}{v - u}. \tag{1}$$

We now apply this result to bound the approximation error on each pieces of the subdivision. Let $k \in [m]$. Recall $\hat{f}$ is linear on the subdivision $[x_k, x_{k+1}]$ and $\hat{f}(x_k) = f(x_k)$ and $\hat{f}(x_{k+1}) = f(x_{k+1})$. Hence, for all $x \in [x_k, x_{k+1}]$, $\hat{f}(x) = f(x_k) + (x - x_k)\frac{f(x_{k+1}) - f(x_k)}{x_{k+1} - x_k}$. Using Equation equation 1 with $u = x_k$ and $v = x_{k+1}$, we get,

$$\|f - \hat{f}\|_{\infty, [x_k, x_{k+1}]} \leq \sup_{x \in [x_k, x_{k+1}]} \left| (x - x_k)(x_{k+1} - x)\frac{\int_{\alpha_x}^{\beta_x} f''(t)dt}{x_{k+1} - x_k} \right|$$

$$\leq \frac{1}{2}(x_{k+1} - x_k)\int_{x_k}^{x_{k+1}} |f''(t)|dt$$

$$\leq \frac{1}{2}(x_{k+1} - x_k)^2 \|f''\|_{\infty, [x_k, x_{k+1}]}.$$

Therefore, using a regular subdivision with step $\sqrt{\frac{2\epsilon}{\|f''\|_\infty}}$ yields an $\epsilon$-approximation of $f$ with $\left\lceil \frac{(b-a)\sqrt{\|f''\|_\infty}}{\sqrt{2\epsilon}} \right\rceil$ pieces.

We now show that for any $\mu > 0$, there exists an $\epsilon$-approximation of $f$ with at most $\frac{\int \sqrt{|f''|}}{\sqrt{2\epsilon}}(1 + \mu)$ pieces. To do so, we use the fact that the upper Riemann sum for $\sqrt{f''}$ converges to the integral since $\sqrt{f''}$ is continuous on $[a, b]$. First define a partition $a = X_0 \leq X_K = b$ of $[a, b]$ such that the upper Riemann sum $\mathcal{R}(\sqrt{f''})$ on this subdivision satisfies $\mathcal{R}(\sqrt{f''}) \leq (1 + \mu/2)\int_a^b \sqrt{f''}$. Now define on each interval $I_k$ of the partition a regular subdivision with step $\sqrt{\frac{2\epsilon}{\|f''\|_{I_k}}}$ as before. Finally, consider the subdivision union of all these subdivisions, and construct the approximation $\hat{f}$ on this final subdivision. By construction, $\|f - \hat{f}\|_\infty \leq \epsilon$ because the inequality holds on each piece of the subdivision. Further, the number of pieces is

$$\sum_{i=0}^{K-1} 1 + \frac{(X_{i+1} - X_i)\sup_{[X_i, X_{i+1}]}\sqrt{f''}}{\sqrt{2\epsilon}} = \frac{\mathcal{R}(\sqrt{f''})}{\sqrt{2\epsilon}} + K \leq \frac{\int \sqrt{|f''|}}{\sqrt{2\epsilon}}(1 + \mu),$$

for $\epsilon > 0$ small enough. Using Lemma B.1 we can complete the proof. $\square$

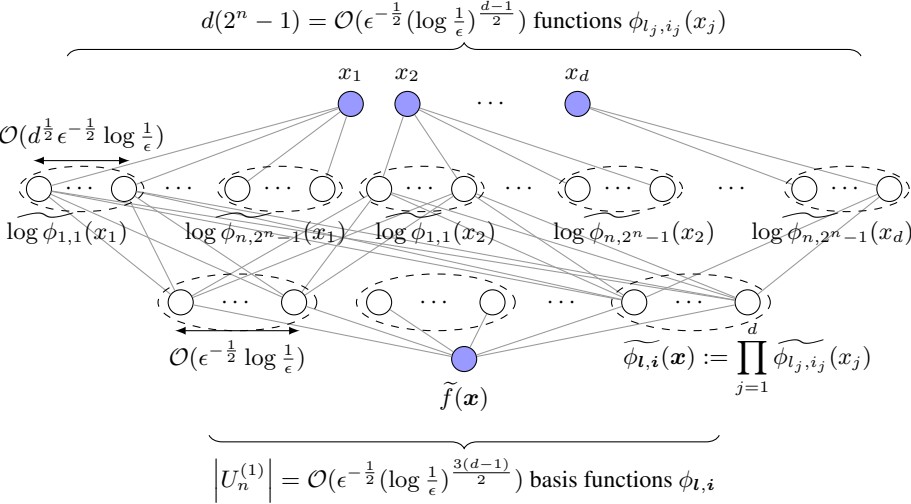

Figure 5: Shallow neural network with ReLU activation approximating a Korobov function $f \in X^{2,\infty}(\Omega)$ within $\epsilon$ in infinity norm. The network has $\mathcal{O}(\epsilon^{-1}(\log \frac{1}{\epsilon})^{\frac{d+1}{2}})$ neurons on the first layer and $\mathcal{O}(\epsilon^{-1} \left(\log \frac{1}{\epsilon}\right)^{\frac{3(d-1)}{2}+1})$ neurons on the second layer.

### B.2.2 PROOF OF COROLLARY B.4

*Proof.* In view of Lemma B.3, the goal is to show that we can remove the dependence of $\mu(f, \epsilon)$ in $\delta$. This essentially comes from the fact that the upper Riemann sum behaves well for approximating log. Consider the subdivision $x_0 := \delta \leq x_1 \leq \ldots \leq x_m \leq x_{m+1} := 1$ with $m = \lfloor \frac{1}{\tilde{\epsilon}} \log \frac{1}{\delta} \rfloor$ where $\tilde{\epsilon} := \log(1 + \sqrt{2\epsilon})$, such that $x_k = e^{\log \delta + k\tilde{\epsilon}}$, for $k = 0, \ldots, m - 1$. Denote $\hat{f}$ the corresponding piece-wise affine approximation. Similarly to the proof of Lemma B.3, for $k = 0, \ldots, m - 1$,

$$\| \log - \hat{f}\|_{\infty,[x_k,x_{k+1}]} \leq \frac{1}{2}(x_{k+1} - x_k)^2 \|f''\|_{\infty,[x_k,x_{k+1}]} \leq \frac{(e^{\tilde{\epsilon}} - 1)^2}{2} \leq \epsilon.$$

The proof follows. $\qquad\square$

### B.3 PROOF OF THEOREM 3.1: APPROXIMATING THE KOROBOV SPACE $X^{2,\infty}(\Omega)$

In this subsection, we prove Theorem 3.1 and show that we can approximate any Korobov function $f \in X^{2,\infty}(\Omega)$ within $\epsilon$ with a 2-layer neural network of $\mathcal{O}(\epsilon^{-\frac{1}{2}}(\log \frac{1}{\epsilon})^{\frac{3(d-1)}{2}})$ neurons. We illustrate the construction in Figure 5. Our proof combines the constructed network approximating the product function and a decomposition of $f$ as a sum of separable functions, i.e. a decomposition of the form

$$f(\boldsymbol{x}) \approx \sum_{k=1}^{K} \prod_{j=1}^{d} \phi_j^{(k)}(x_j), \quad \forall \boldsymbol{x} \in [0, 1]^d.$$

Consider the sparse grid construction of the approximating space $V_n^{(1)}$ using the standard hat function as mother function to create the hierarchical basis $W_{\boldsymbol{l}}$ (introduced in Section 2.2). We recall that the approximation space is defined as $V_n^{(1)} := \bigoplus_{|\boldsymbol{l}|_1 \leq n+d-1} W_{\boldsymbol{l}}$. We will construct a neural network approximating the sparse grid approximation and then use the result of Theorem 2.2 to derive the approximation error. Figure 5 gives an illustration of the construction.

Let $f_n^{(1)}$ be the projection of $f$ on the subspace $V_n^{(1)}$. $f_n^{(1)}$ can be written as

$$f_n^{(1)}(\boldsymbol{x}) = \sum_{(\boldsymbol{l},\boldsymbol{i}) \in U_n^{(1)}} v_{\boldsymbol{l},\boldsymbol{i}} \phi_{\boldsymbol{l},\boldsymbol{i}}(\boldsymbol{x}),$$

where $U_n^{(1)}$ contains the indices $(\boldsymbol{l}, \boldsymbol{i})$ of basis functions present in $V_n^{(1)}$ i.e.

$$U_n^{(1)} := \{(\mathbf{l}, \mathbf{i}), \quad |\mathbf{l}|_1 \leq n + d - 1, \ \mathbf{1} \leq \mathbf{i} \leq 2^{\mathbf{l}} - \mathbf{1}, \ i_j \text{ odd for all } 1 \leq j \leq d\}. \quad (2)$$

Throughout the proof, we explicitly construct a neural network that uses this decomposition to approximate $f_n^{(1)}$. We then use Theorem 2.2 and choose $n$ carefully such that $f_n^{(1)}$ approximates $f$ within $\epsilon$ for $L^\infty$ norm. Note that the basis functions can be written as a product of univariate functions $\phi_{\boldsymbol{l}, \boldsymbol{i}} = \prod_{j=1}^d \phi_{l_j, i_j}$. We can therefore use the product approximation of Proposition 3.2 to approximate the basis functions. Specifically, we will use one layer to approximate the terms $\log \phi_{l_j, i_j}$ and a second layer to approximate the exponential.

We now present in detail the construction of the first layer. First, recall that $\phi_{l_j, i_j}$ is a piece-wise affine function with subdivision $0 \leq \frac{i_j - 1}{2^{l_j}} \leq \frac{i_j}{2^{l_j}} \leq \frac{i_j + 1}{2^{l_j}} \leq 1$. Define the error term $\tilde{\epsilon} := \frac{\epsilon}{2|f|_{2,\infty}}$. We consider a symmetric subdivision of the interval $\left[\frac{i_j - 1 + \tilde{\epsilon}}{2^{l_j}}, \frac{i_j + 1 - \tilde{\epsilon}}{2^{l_j}}\right]$. We define it as follows: $x_0 = \frac{i_j - 1 + \tilde{\epsilon}}{2^{l_j}} \leq x_1 \leq \cdots \leq x_{m+1} = \frac{i_j}{2^{l_j}} \leq x_{m+2} \leq \cdots \leq x_{2m+2} = \frac{i_j + 1 - \tilde{\epsilon}}{2^{l_j}}$ where $m = \left\lfloor \frac{1}{\epsilon_0} \log \frac{1}{\tilde{\epsilon}} \right\rfloor$ and $\epsilon_0 := \log(1 + \sqrt{2\tilde{\epsilon}/d})$, such that

$$x_k = \frac{i_j - 1 + e^{\log \tilde{\epsilon} + k\epsilon_0}}{2^{l_j}} \qquad 0 \leq k \leq m,$$

$$x_k = \frac{i_j + 1 - e^{\log \tilde{\epsilon} + (2m+2-k)k\epsilon_0}}{2^{l_j}} \quad m + 2 \leq k \leq 2m + 2.$$

Note that with this definition, the terms $\log(2^{l_j} x_k - i_j)$ form a regular sequence with step $\epsilon_0$. We now construct the piece-wise affine function $\hat{g}_{l_j, i_j}$ on the subdivision $x_0 \leq \cdots \leq x_{2m+2}$ which coincides with $\log \phi_{l_j, i_j}$ on $x_0, \cdots, x_{2m+2}$ and is constant on $[0, x_0]$ and $[x_{2m+2}, 1]$. By Lemma B.1, this function can be represented by a 1-layer neural network with as much neurons as the number of pieces of $\hat{g}$, i.e. at most $2\sqrt{\frac{3d}{\tilde{\epsilon}}} \log \frac{1}{\tilde{\epsilon}}$ neurons for $\epsilon$ sufficiently small. A similar proof to that of Corollary B.4 shows that $\hat{g}$ approximates $\max(\log \phi_{l_j, i_j}, \log(\tilde{\epsilon}/3))$ within $\tilde{\epsilon}/(3d)$ for the infinity norm. We use this construction to compute in parallel, $\tilde{\epsilon}/(3d)$-approximations of $\max(\log \phi_{l_j, i_j}(x_j), \log \tilde{\epsilon})$ for all $1 \leq j \leq d$, and $1 \leq l_j \leq n, 1 \leq i_j \leq 2^{l_j}$ where $i_j$ is odd. These are exactly the 1-dimensional functions that we will need, in order to compute the $d$-dimensional function basis of the approximation space $V_n^{(1)}$. There are $d(2^n - 1)$ such univariate functions, therefore our first layer contains at most $2^{n+1} d \sqrt{\frac{3d}{\tilde{\epsilon}}} \log \frac{1}{\tilde{\epsilon}}$ neurons.

We now turn to the second layer. The result of the first two layers will be $\tilde{\epsilon}/3$-approximations of $\phi_{\mathbf{l}, \mathbf{i}}$ for all $(\mathbf{l}, \mathbf{i}) \in U_n^{(1)}$. Recall that $U_n^{(1)}$ contains the indices for the functions forming a basis of the approximation space $V_n^{(1)}$. To do so, for each indexes $(\mathbf{l}, \mathbf{i}) \in U_n^{(1)}$ we construct a 1-layer neural network approximating the function $\exp$, which will compute an approximation of $\exp(\hat{g}_{l_1, i_1} + \cdots + \hat{g}_{l_d, i_d})$. The approximation of $\exp$ is constructed in the same way as for Lemma B.3. Consider a regular subdivision of the interval $[\log(\tilde{\epsilon}/3), 0]$ with step $\sqrt{2(\tilde{\epsilon}/3)}$, i.e. $x_0 := \log(\tilde{\epsilon}/3) \leq x_1 \leq \cdots \leq x_m \leq x_{m+1} = 0$ where $m = \left\lfloor \sqrt{\frac{3}{2\tilde{\epsilon}}} \log \frac{3}{\tilde{\epsilon}} \right\rfloor$, such that $x_k = \log \tilde{\epsilon} + k\sqrt{2\tilde{\epsilon}}, \quad 0 \leq k \leq m$. Construct the piece-wise affine function $\hat{h}$ on the subdivision $x_0 \leq \cdots \leq x_{m+1}$ which coincides with $\exp$ on $x_0, \cdots, x_{m+1}$ and is constant on $(-\infty, x_0]$. Lemma B.3 shows that $\hat{h}$ approximates $\exp$ on $\mathbb{R}_-$ within $\tilde{\epsilon}$ for the infinity norm. Again, Lemma B.1 gives a representation of $\hat{h}$ as a 1-layer neural network with as many neurons as pieces in $\hat{h}$ i.e. $1 + \left\lceil \sqrt{\frac{3}{2\tilde{\epsilon}}} \log \frac{3}{\tilde{\epsilon}} \right\rceil$. The second layer is the union of 1-layer neural networks approximating $\exp$ within $\tilde{\epsilon}/3$, for each indexes $(\mathbf{l}, \mathbf{i}) \in U_n^{(1)}$. Therefore, the second layer contains $\left|U_n^{(1)}\right| \left(1 + \left\lceil \sqrt{\frac{3}{2\tilde{\epsilon}}} \log \frac{3}{\tilde{\epsilon}} \right\rceil\right)$ neurons. As shown in Bungartz & Griebel (2004),

$$\left|U_n^{(1)}\right| = \sum_{i=0}^{n-1} 2^i \cdot \binom{d-1+i}{d-1} = (-1)^d + 2^n \sum_{i=0}^{d-1} \binom{n+d-1}{i}(-2)^{d-1-i} = 2^n \cdot \left(\frac{n^{d-1}}{(d-1)!} + \mathcal{O}(n^{d-2})\right).$$

Therefore, the second layer has $O\left(2^n \frac{n^{d-1}}{(d-1)!} \tilde{\epsilon}^{-\frac{1}{2}} \log \frac{1}{\tilde{\epsilon}}\right)$ neurons. Finally, the output layer computes the weighted sum of the basis functions to approximate $f_n^{(1)}$. Denote by $\hat{f}_n^{(1)}$ the corresponding function of the constructed neural network (see Figure 5), i.e.

$$\hat{f}_n^{(1)} = \sum_{(\mathbf{l},\mathbf{i}) \in U_n^{(1)}} v_{\mathbf{l},\mathbf{i}} \cdot \hat{h}\left(\sum_{j=1}^d \hat{g}(x_j)\right).$$

Let us analyze the approximation error of our neural network. The proof of Proposition 3.2 shows that the output of the two first layers $h\left(\sum_{j=1}^d \hat{g}(\cdot_j)\right)$ approximates $\phi_{\mathbf{l},\mathbf{i}}$ within $\tilde{\epsilon}$. Therefore, we obtain $\|f_n^{(1)} - \hat{f}_n^{(1)}\|_\infty \leq \tilde{\epsilon} \sum_{(\mathbf{l},\mathbf{i}) \in U_n^{(1)}} |v_{\mathbf{l},\mathbf{i}}|$. We now use approximation bounds from Theorem 2.2 on $f_n^{(1)}$.

$$\|f - \hat{f}_n^{(1)}\|_\infty \leq \|f - f_n^{(1)}\|_\infty + \|f_n^{(1)} - \hat{f}_n^{(1)}\|_\infty \leq \frac{2 \cdot |f|_{2,\infty}}{8^d} \cdot 2^{-2n} \cdot A(d,n) + \frac{\epsilon}{2|f|_{2,\infty}} \sum_{(\mathbf{l},\mathbf{i}) \in U_n^{(1)}} |v_{\mathbf{l},\mathbf{i}}|,$$

where

$$\sum_{(\mathbf{l},\mathbf{i}) \in U_n^{(1)}} |v_{\mathbf{l},\mathbf{i}}| \leq |f|_{2,\infty} 2^{-d} \sum_{i \geq 0} 2^{-i} \cdot \binom{d-1+i}{d-1} \leq |f|_{2,\infty}.$$

Let us now take $n_\epsilon = \min\left\{n : \frac{2|f|_{2,\infty}}{8^d} 2^{-2n} A(d,n) \leq \frac{\epsilon}{2}\right\}$. Then, using the above inequality shows that the neural network $\hat{f}_{n_\epsilon}^{(1)}$ approximates $f$ within $\epsilon$ for the infinity norm. We will now estimate the number of neurons in each layer of this network. Note that

$$n_\epsilon \sim \frac{1}{2\log 2} \log \frac{1}{\epsilon} \quad \text{and} \quad 2^{n_\epsilon} \leq \frac{4}{8^{\frac{d}{2}}(2\log 2)^{\frac{d-1}{2}}(d-1)!^{\frac{1}{2}}} \sqrt{\frac{|f|_{2,\infty}}{\epsilon}} \left(\log \frac{1}{\epsilon}\right)^{\frac{d-1}{2}} \cdot (1 + \mathcal{O}(1)).$$
$$(3)$$

We can use the above estimates to show that the constructed neural network has at most $N_1$ (resp. $N_2$) neurons on the first (resp. second) layer where

$$N_1 \underset{\epsilon \to 0}{\sim} \frac{8\sqrt{6}d^2}{8^{\frac{d}{2}}(2\log 2)^{\frac{d-1}{2}}d!^{\frac{1}{2}}} \cdot \frac{|f|_{2,\infty}}{\epsilon} \left(\log \frac{1}{\epsilon}\right)^{\frac{d+1}{2}},$$

$$N_2 \underset{\epsilon \to 0}{\sim} \frac{4\sqrt{3}d^{\frac{3}{2}}}{8^{d/2}(2\log 2)^{\frac{3(d-1)}{2}}d!^{\frac{3}{2}}} \cdot \frac{|f|_{2,\infty}}{\epsilon} \left(\log \frac{1}{\epsilon}\right)^{\frac{3(d-1)}{2}+1}.$$

This proves the bound the number of neurons. Finally, to prove the bound on the number of training parameters of the network, notice that the only parameters of the network that depend on the function $f$ are the parameters corresponding to the weighs $v_{l,i}$ of the sparse grid decomposition. This number is $|U_{n_\epsilon}^{(1)}| = \mathcal{O}(2^{n_\epsilon} n_\epsilon^{d-1}) = \mathcal{O}(\epsilon^{-\frac{1}{2}} (\log \frac{1}{\epsilon})^{\frac{3(d-1)}{2}})$.

### B.4 PROOF OF THEOREM 3.4: GENERALIZATION TO GENERAL ACTIVATION FUNCTIONS

We start by formalizing the intuition that a sigmoid-like (resp. ReLU-like) function is a function that resembles the Heaviside (resp. ReLU) function by zooming out along the $x$ (resp. $x$ and $y$) axis.

**Lemma B.6.** *Let $\sigma$ be a sigmoid-like activation with limit $a$ (resp. $b$) in $-\infty$ (resp. $+\infty$). For any $\delta > 0$ and error tolerance $\epsilon > 0$, there exists a scaling $M > 0$ such that $x \mapsto \frac{\sigma(Mx)}{b-a} - a$ approximates the Heaviside function within $\epsilon$ outside of $(-\delta, \delta)$ for the infinity norm. Furthermore, this function has values in $[0, 1]$.*

*Let $\sigma$ be a ReLU-like activation with asymptote $b \cdot x + c$ in $+\infty$. For any $\delta > 0$ and error tolerance $\epsilon > 0$, there exists a scaling $M > 0$ such that $x \mapsto \frac{\sigma(Mx)}{Mb}$ approximates the ReLU function within $\epsilon$ for the infinity norm.*

*Proof.* Let $\delta, \epsilon > 0$ and $\sigma$ a sigmoid-like activation with limit $a$ (resp. $b$) in $-\infty$ (resp. $+\infty$). There exists $x_0 > 0$ sufficiently large such that $(b-a)|\sigma(x)-a| \leq \epsilon$ for $x \leq -x_0$ and $(b-a)|\sigma(x)-b| \leq \epsilon$ for $x \geq x_0$. It now suffices to take $M := x_0/\delta$ to obtain the desired result.

Now let $\sigma$ be a ReLU-like activation with oblique asymptote $bx$ in $+\infty$ where $b > 0$. Let $M$ such that $|\sigma| \leq Mb\epsilon$ for $x \leq 0$ and $|\sigma(x) - bx| \leq Mb\epsilon$ for $x \geq 0$. One can check that $|\frac{\sigma(Mx)}{Mb}| \leq \epsilon$ for $x \leq 0$, and $|\frac{\sigma(Mx)}{Mb} - x| \leq \epsilon$ for $x \geq 0$. $\qquad\square$

Using this approximation, we reduce the analysis of sigmoid-like (resp. ReLU-like) activations to the case of a Heaviside (resp ReLU) activation to prove the desired theorem.

**Proof of Theorem 3.4** We start by the class of ReLU-like activations. Let $\sigma$ be a ReLU-like activation function. Lemma B.6 shows that one can approximate arbitrarily well the ReLU activation with a linear map $\sigma$. Take the neural network approximator $\hat{f}$ of a target function $f$ given by Theorem 3.1. At each node, we can add the linear map corresponding to $x \mapsto \frac{\sigma(Mx)}{Mb}$ with no additional neuron nor parameter. Because the approximation is continuous, we can take $M > 0$ arbitrarily large in order to approximate $\hat{f}$ with arbitrary precision on the compact $[0,1]^d$.

The same argument holds for sigmoid-like activation functions in order to reduce the problem to Heaviside activation functions. Although quadratic approximations for univariate functions similar to Lemma B.3 are not valid for general sigmoid-like activations – in particular the Heaviside — we can obtain an analog to Lemma B.2 as Lemma B.7 given in the Appendix B.4.1. This results is an increased number of neurons. In order to approximate a target function $f \in X^{2,\infty}(\Omega)$, we use the same structure as the neural network constructed for ReLU activations and use the same notations as in the proof of Theorem 3.1. The first difference lies in the approximation of $\log \phi_{l_j,i_j}$ in the first layer. Instead of using Corollary B.4, we use Lemma B.7. Therefore, $\frac{12d}{\tilde{\epsilon}} \log \frac{3}{\tilde{\epsilon}}$ neurons are needed to compute a $\tilde{\epsilon}/(3d)$-approximation of $\max(\log \phi_{l_j,i_j}, \log(\tilde{\epsilon}/3))$. The second difference is in the approximation of the exponential in the second layer. Again, we use Lemma B.7 to construct a $\tilde{\epsilon}/3$-approximation of the exponential on $\mathbb{R}_-$ with $\frac{6}{\tilde{\epsilon}}$ neurons for the second layer. As a result, the first layer contains at most $2^{n+2} \frac{3d^2}{\tilde{\epsilon}} \log \frac{1}{\tilde{\epsilon}}$ neurons for $\epsilon$ sufficiently small, and the second layer contains $\left|U_n^{(1)}\right| \frac{6}{\tilde{\epsilon}}$ neurons. Using the same estimates as in the proof of Theorem 3.1 shows that the constructed neural network has at most $N_1$ (resp. $N_2$) neurons on the first (resp. second) layer where

$$N_1 \underset{\epsilon \to 0}{\sim} \frac{3 \cdot 2^5 \cdot d^{5/2}}{8^{\frac{d}{2}} (2 \log 2)^{\frac{d-1}{2}} d!^{\frac{1}{2}}} \frac{|f|_{2,\infty}^{\frac{3}{2}}}{\epsilon^{\frac{3}{2}}} \left( \log \frac{1}{\epsilon} \right)^{\frac{d+1}{2}},$$

$$N_2 \underset{\epsilon \to 0}{\sim} \frac{24 \cdot d^{\frac{3}{2}}}{8^{\frac{d}{2}} (2 \log 2)^{\frac{3(d-1)}{2}} d!^{\frac{3}{2}}} \cdot \frac{|f|_{2,\infty}^{\frac{3}{2}}}{\epsilon^{\frac{3}{2}}} \left( \log \frac{1}{\epsilon} \right)^{\frac{3(d-1)}{2}}.$$

This ends the proof.

### B.4.1 PROOF OF LEMMA B.7

**Lemma B.7.** *Let $\sigma$ be a sigmoid-like activation. Let $f : I \longrightarrow [c,d]$ be a right-continuous increasing function where $I$ is an interval, and let $\epsilon > 0$. There exists a shallow neural network with activation $\sigma$, with at most $2\frac{d-c}{\epsilon}$ neurons on a single layer, that approximates $f$ within $\epsilon$ for the infinity norm.*

*Proof.* The proof is analog to that of Lemma B.2. Let $m = \lfloor \frac{d-c}{\epsilon} \rfloor$. We define a regular subdivision of the image interval $c \leq y_1 \leq \dots \leq y_m \leq d$ where $y_k = c + k\epsilon$ for $k = 1, \dots, m$, then using the monotony of $f$, we can define a subdivision of $I$, $x_1 \leq \dots \leq x_m$ such that $x_k := \sup\{x \in I, f(x) \leq y_k\}$. Let us first construct an approximation neural network $\hat{f}$ with the Heaviside activation. Consider

$$\hat{f}(x) := y_1 + \epsilon \sum_{i=1}^{m-1} \mathbf{1}\left( x - \frac{x_i + x_{i+1}}{2} \geq 0 \right).$$

Let $x \in [c, d]$ and $k$ such that $x \in [x_k, x_{k+1}]$. We have by monotony $y_k \leq f(x) \leq y_{k+1}$ and $y_k = y_1 + (k-1)\epsilon \leq \hat{f}(x) \leq y_1 + k\epsilon = y_{k+1}$. Hence, $\hat{f}$ approximates $f$ within $\epsilon$ in infinity norm.

Let $\delta < \min_{i=1,\dots,m}(x_{i+1} - x_i)/4$ and $\sigma$ a general sigmoid-like activation with limits $a$ in $-\infty$ and $b$ in $+\infty$. Take $M$ given by Lemma B.7 such that $\frac{\sigma(Mx)}{b-a} - a$ approximates the Heaviside function within $1/m$ outside of $(-\delta, \delta)$ and has values in $[0, 1]$. Using the same arguments as above, the function

$$\hat{f}(x) := y_1 + \epsilon \sum_{i=1}^{m-1} \frac{\sigma\left(Mx - M\frac{x_i + x_{i+1}}{2}\right)}{b - a} - a$$

approximates $f$ within $2\epsilon$ for the infinity norm. The proof follows. □

## C  PROOFS OF SECTION 4

### C.1  PROOF OF THEOREM 4.1: APPROXIMATING KOROBOV FUNCTIONS WITH DEEP NEURAL NETWORKS

Let $\epsilon > 0$. We construct a similar structure to the network defined in Theorem 3.1 by using the sparse grid approximation of Subsection 2.2. For a given $n$, let $f_n^{(1)}$ be the projection of $f$ in the approximation space $V_n^{(1)}$ (defined in Subsection 2.2) and $U_n^{(1)}$ (defined in equation 2) the set of indices $(l, i)$ of basis functions present in $V_n^{(1)}$. Recall $f_n^{(1)}$ can be uniquely decomposed as

$$f_n^{(1)}(\mathbf{x}) = \sum_{(\mathbf{l},\mathbf{i}) \in U_n^{(1)}} v_{\mathbf{l},\mathbf{i}} \phi_{\mathbf{l},\mathbf{i}}(\mathbf{x}).$$

where $\phi_{l,i} = \prod_{j=1}^{d} \phi_{l_j, i_j}$ are the basis functions defined in Subsection 2.2. In the first layer, we compute exactly the piece-wise linear hat functions $\phi_{l_j, i_j}$, then in the next set of layers, we use the product-approximating neural network given by Proposition 4.2 to compute the basis functions $\phi_{l,i} = \prod_{j=1}^{d} \phi_{l_j, i_j}$ (see Figure 3). The output layer computes the weighted sum $\sum_{(\mathbf{l},\mathbf{i}) \in U_n^{(1)}} v_{\mathbf{l},\mathbf{i}} \phi_{\mathbf{l},\mathbf{i}}(\mathbf{x})$ and outputs $f_n^{(1)}$. Because the approximation has arbitrary precision, we can chose the network of Proposition 4.2 such that the resulting network $\hat{f}$ verifies $\|\hat{f} - f_n^{(1)}\|_\infty \leq \epsilon/2$.

More precisely, as $\phi_{l_j, i_j}$ is piece-wise linear with four pieces, we can compute it exactly with four neurons with ReLU activation on a single layer (Lemma B.1). Our first layer is composed of the union of all these ReLU neurons, for the $d(2^n - 1)$ indices $l_j, i_j$ such that $1 \leq j \leq d$, $1 \leq l_j \leq n$, $1 \leq i_j \leq 2^{l_j}$ and $i_j$ is odd. Therefore, it contains at most $d2^{n+2}$ neurons with ReLU activation. The second set of layers is composed of the union of product-approximating neural networks to compute $\phi_{l,i}$ for all $(l, i) \in U_n^{(1)}$. This set of layers contains $\lceil \log_2 d \rceil$ layers with activation $\sigma$ and at most $|U_n^{(1)}| \cdot 8d$ neurons. The output of these two sets of layers is an approximation of the basis functions $\phi_{l,i}$ with arbitrary precision. Consequently, the final output of the complete neural network is an approximation of $f_n^{(1)}$ with arbitrary precision. Similarly to the proof of Theorem 3.1, we can chose the smallest $n$ such that $\|f - f_n^{(1)}\|_\infty \leq \epsilon/2$ (see equation 3 for details). Finally, the network has depth at most $\log_2 d + 2$ and $N$ neurons where

$$N = 8d|U_{n_\epsilon}^{(1)}| \underset{\epsilon \to 0}{\sim} \frac{2^5 \cdot d^{5/2}}{8^{\frac{d}{2}}(2\log 2)^{\frac{3(d-1)}{2}} d!^{\frac{3}{2}}} \cdot \sqrt{\frac{|f|_{2,\infty}}{\epsilon}} \left(\log \frac{1}{\epsilon}\right)^{\frac{3(d-1)}{2}}.$$

The parameters of the network depending on the function are exactly the coefficients $v_{l,i}$ of the sparse grid approximation. Hence, the network has $\mathcal{O}(\epsilon^{-\frac{1}{2}}(\log \frac{1}{\epsilon})^{\frac{3(d-1)}{2}})$ training parameters.

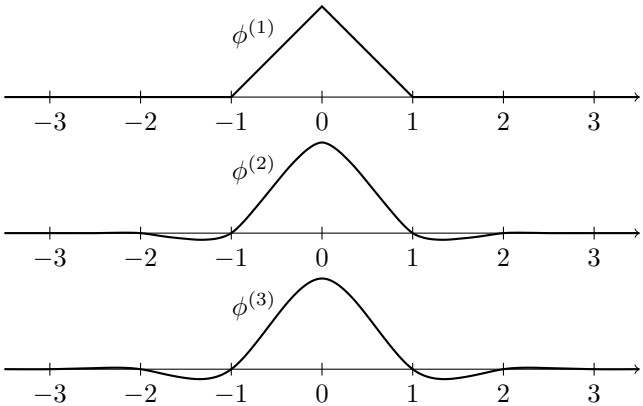

Figure 6: Deslaurier-Dubuc interpolets of degree 1, 2 and 3.

# D PROOFS OF SECTION 5

## D.1 PROOF OF THEOREM 5.2: NEAR-OPTIMALITY OF NEURAL NETWORKS FOR KOROBOV FUNCTIONS

Our goal is to define an appropriate subspace $X_{N+1}$ in order to get a good lower bound on the Bernstein width $b_N(K)_{\mathcal{X}}$, defined in equation 5, which in turn provides a lower bound on the approximation error (Theorem 5.1).

To do so, we introduce the Deslaurier-Dubuc interpolet $\phi^{(L)} : \mathbb{R} \to \mathbb{R}$. The construction of this function uses an interpolating scheme on binary rationals. First, $\phi^{(L)}$ is defined on all integers $\phi^{(L)}(k) = \mathbf{1}_{k=0}$ for $k \in \mathbb{Z}$. Then, we define the function on half integers $\frac{2k+1}{2}$ by fitting a polynomial of degree $2L - 1$ interpolating the standard hat function $\phi$ at $k - L + 1, \cdots, k + L$. Iteratively, we define the interpolet on binary rationals of the form $\frac{2k+1}{2^{j+1}}$ from the value of the interpolet on rationals with denominator $2^j$. Specifically, let $P_{j,k}^{(L)}$ be the unique polynomial of degree $2L$ interpolating $\phi^{(L)}$ at points $\frac{k'}{2^j}$ for $k' \in \{k - L + 1, \cdots, k + L\}$. We set

$$\phi^{(L)}\left(\frac{2k+1}{2^{j+1}}\right) := P_{j,k}^{(L)}\left(\frac{2k+1}{2^j}\right).$$

For example, for $L = 2$ we get $\phi^{(2)}(\frac{2k+1}{2^{j+1}}) := \frac{9}{16}\phi^{(2)}(\frac{k}{2^j}) + \frac{9}{16}\phi^{(2)}(\frac{k+1}{2^j}) - \frac{1}{16}\phi^{(2)}(\frac{k-2}{2^j}) - \frac{1}{16}\phi^{(2)}(\frac{k+3}{2^j})$. This process defines the interpolet on binary rationals. We then extend the function to the real line by continuity. See Figure 6 for an illustration. Deslauriers & Dubuc (1989) proved that the regularity of the interpolet is an increasing function of $L$ the degree of interpolation. We now prove results in the case $L = 2$.

**Lemma D.1.** *The interpolet of degree 2, $\phi^{(2)}$ is $\mathcal{C}^2$ and has support $Supp\left(\phi^{(2)}\right) = [-3, 3]$.*

*Proof.* See Appendix D.2.1. $\qquad\square$

We will now use the interpolet $\phi^{(2)}$ to construct the subspace $X_{N+1}$. Using the sparse grids approach, we can construct a hierarchical basis in $X^{2,\infty}(\Omega)$ using the interpolet $\phi^{(2)}$ as mother function. In the remaining of the proof, we will write $\phi$ instead of $\phi^{(2)}$ for simplicity, and use the notations and definitions related to sparse grids, introduced in Subsection 2.2. Because $E_N(K)$ is decreasing in $N$, it suffices to show the result for $N_n$ when we define our space $X_{N+1}$ to be exactly the approximating space $V_n^{(1)}$ of sparse grids $X_{N_n+1} := V_n^{(1)}$. The following equation establishes the relation between $n$ and $N_n$. In the following, for simplicity, we will write $N$ instead of $N_n$.

$$N = dim(V_n^{(1)}) - 1 = \sum_{|\mathbf{l}|_1 \leq n+d-1} 2^{|\mathbf{l}|_1 - \mathbf{1}} - 1 = \sum_{i=0}^{n-1} 2^{i-d}\binom{d-1+i}{d-1} - 1 = 2^n \cdot \left(\frac{n^{d-1}}{(d-1)!} + \mathcal{O}(n^{d-2})\right).$$

First, let us give some properties about the subspace $X_{N+1}$.

**Proposition D.2.** *Let $u \in X_{N+1}$ and write it in decomposed form $u = \sum_{l,i} v_{l,i} \cdot \phi_{l,i}$, where the sum is taken over all multi-indices corresponding to basis functions of $X_{N+1}$. The coefficients $v_{l,i}$ can be computed in the following way.*

$$v_{l,i} = \left( \prod_{j=1}^{d} I_{l_j, i_j} \right) u =: I_{l,i} u$$

*where $I_{l_j, i_j} u = u(\frac{i_j}{2^j}) - \frac{9}{16} u(\frac{i_j - 1}{2^j}) - \frac{9}{16} u(\frac{i_j + 1}{2^j}) + \frac{1}{16} u(\frac{i_j - 3}{2^j}) + \frac{1}{16} u(\frac{i_j + 3}{2^j})$. Here, $I_{l,i}$ denotes the d-dimensional stencil which gives a linear combination of values of $u$ at $5^d$ nodal points.*

*Proof.* See Appendix D.2.2. □

Note that the stencil representation of the coefficients gives directly $|v_{l,i}| \leq 5^d \|u\|_\infty$. We are now ready to make our estimates. The goal is to compute $\sup\{\rho : \rho U(X_{N+1}) \subset K\}$, which will lead to a bound on $b_N(K)_{\mathcal{X}}$. To do so, it suffices to upper bound the Korobov norm by the $L^\infty$ norm for elements of $X_{N+1}$. In fact, if $\Gamma_d > 0$ satisfies for all $u \in U(X_{N+1})$, $|u|_{X^{2,\infty}} \leq \Gamma_d \|u\|_\infty$, then $b_N(K)_{\mathcal{X}} \geq 1/\Gamma_d$.

Now take $u \in X_{N+1}$ and let us write $u = \sum_{l,i} v_{l,i} \cdot \phi_{l,i}$. Note that basis functions in the same hierarchical class $W_l$ are almost disjoint. More precisely, at each point $x \in \Omega$, at most $3^d$ basis functions $\phi_{l,i}$ have non-zero contribution to $u(x)$. Therefore, for any $0 \leq \alpha \leq 2 \cdot 1$,

$$\|D^\alpha u\|_\infty \leq \sum_{|l|_1 \leq n+d-1} 3^d \max_{1 \leq i \leq 2^l - 1, \ i \ \text{odd}} |v_{l,i}| \cdot 2^{\langle \alpha, l \rangle} \|D^\alpha \phi\|_\infty$$

$$= 60^d |\phi|_{X^{2,\infty}(\Omega)} \|u\|_\infty \cdot \sum_{i=0}^{n-1} 2^{2i} \binom{d-1+i}{d-1}$$

$$= \frac{60^d}{(d-1)!} |\phi|_{X^{2,\infty}(\Omega)} \cdot 2^{2n} \left( n^{d-1} + \mathcal{O}(n^{d-2}) \right) \|u\|_\infty.$$

Finally, denoting by $C_d$ the constant $\frac{60^d}{(d-1)!} |\phi|_{X^{2,\infty}(\Omega)}$, we get for $n$ sufficiently large

$$b_N(K)_{\mathcal{X}} \geq \frac{1}{2C_d} \cdot \frac{1}{2^{2n} \cdot n^{d-1}}.$$

Furthermore, recall $N = \frac{1}{(d-1)!} 2^n \cdot n^{d-1} \cdot \left( 1 + O\left(\frac{1}{n}\right) \right)$. Therefore, $n \sim \frac{\log N}{\log 2}$, and $2^n \sim (d-1)!(\log 2)^{d-1} \cdot \frac{N}{(\log N)^{d-1}}$. Finally we obtain for some constant $c_d > 0$,

$$b_N(K)_{\mathcal{X}} \geq c_d \frac{1}{N^2} (\log N)^{d-1}.$$

We conclude by analyzing the minimum number of parameters in order to get an $\epsilon$-approximation of the Korobov unit ball $K$. Define $n_\epsilon := \min\{n : \frac{2}{C_d} \frac{1}{2^{2n} \cdot n^{d-1}} \leq \epsilon\}$. This yields $n_\epsilon \sim \frac{1}{2 \log 2} \log \frac{1}{\epsilon}$, and $2^n \sim \sqrt{\frac{2}{C_d}} (2 \log 2)^{d-1} \frac{1}{\sqrt{\epsilon} \cdot (\log \frac{1}{\epsilon})^{\frac{d-1}{2}}}$. The number of needed parameters to obtain an $\epsilon$ approximation $K$ is therefore

$$N_\epsilon \sim \tilde{C}_d \cdot \frac{1}{\sqrt{\epsilon}} \left( \log \frac{1}{\epsilon} \right)^{\frac{d-1}{2}},$$

for some constant $\tilde{C}_d > 0$.

Interestingly, the subspace $X_{N+1}$ our proof uses to show the lower bound is essentially the same as the subspace we use to approximate Koborov functions in our proof of the upper bounds (Theorem 3.1, 4.1). The difference is in the choice of the interpolate $\phi$ to construct the basis functions, degree 1 for the former, and 2 for the later.

## D.2 MISSING PROOFS OF SUBSECTION D.1

### D.2.1 PROOF OF LEMMA D.1

**Lemma D.3.** *The interpolet of degree 2, $\phi^{(2)}$ is $\mathcal{C}^2$ and has support $Supp\left(\phi^{(2)}\right) = [-3,3]$.*

*Proof.* To analyze the regularity of the interpolet, we introduce the trigonometric polynomial

$$P^{(2)}(\theta) := \sum_{k \in \mathbb{Z}} \phi^{(2)}\left(\frac{k}{2}\right) e^{ik\theta} = 1 + \frac{9}{16}(e^{i\theta} + e^{-i\theta}) - \frac{1}{16}(e^{3i\theta} + e^{-3i\theta}).$$

We can write $P^{(2)}(\theta)$ as

$$P^{(2)}(\theta) = \left[\frac{\sin\theta}{\sin\left(\frac{\theta}{2}\right)}\right]^4 \cdot S(\theta),$$

where $S(\theta) = \frac{1}{4} - \frac{1}{16}(e^{i\theta} + e^{-i\theta})$ is a trigonometric polynomial of degree 1. Deslaurier and Dubuc (Deslauriers & Dubuc, 1989, Theorem 7.11) showed that the interpolet $\phi^{(2)}$ has regularity $\left\lfloor -\frac{\log r}{\log 2} \right\rfloor$ where $r$ is the spectral radius of the matrix $B := [s_{j-2k}]_{-1 \leq j,k \leq 1}$ where $s_j$ are the coefficients of the trigonometric polynomial $S(\theta)$. In our case, matrix $B$ writes

$$B = \begin{pmatrix} -1/16 & -1/16 & 0 \\ 0 & 1/4 & 0 \\ 0 & -1/16 & -1/16 \end{pmatrix},$$

and has spectral radius $r = 1/4$. Therefore, the regularity of $\phi^{(2)}$ is at least $\left\lfloor -\frac{\log r}{\log 2} \right\rfloor = 2$. For the support, one can check that more generally $Supp(\phi^{(L)}) = [-2L+1, 2L-1]$. $\qquad\square$

### D.2.2 PROOF OF PROPOSITION D.2

**Proposition D.4.** *Let $u \in X_{N+1}$ and write it in decomposed form $u = \sum_{\boldsymbol{l},\boldsymbol{i}} v_{\boldsymbol{l},\boldsymbol{i}} \cdot \phi_{\boldsymbol{l},\boldsymbol{i}}$, where the sum is taken over all multi-indices corresponding to basis functions of $X_{N+1}$. The coefficients $v_{\boldsymbol{l},\boldsymbol{i}}$ can be computed in the following way.*

$$v_{\boldsymbol{l},\boldsymbol{i}} = \left(\prod_{j=1}^{d} I_{l_j,i_j}\right) u =: I_{\boldsymbol{l},\boldsymbol{i}} u$$

*where $I_{l_j,i_j} u = u(\frac{i_j}{2^j}) - \frac{9}{16} u(\frac{i_j-1}{2^j}) - \frac{9}{16} u(\frac{i_j+1}{2^j}) + \frac{1}{16} u(\frac{i_j-3}{2^j}) + \frac{1}{16} u(\frac{i_j+3}{2^j})$. Here, $I_{\boldsymbol{l},\boldsymbol{i}}$ denotes the $d$-dimensional stencil which gives a linear combination of values of $u$ at $5^d$ nodal points.*

*Proof.* We start by looking at the 1-dimensional case. One can check that $I_{l,i}\phi_{\tilde{l},\tilde{i}} = \mathbf{1}_{\tilde{l}=l,\tilde{i}=i}$. Indeed, if $\tilde{l} > l$ of if $\tilde{l} = l$ and $\tilde{i} \neq i$, then $\phi_{\tilde{l},\tilde{i}}$ will be zero at the nodal values of $I_{l,i}$. Further, if $\tilde{l} < l$ the iterative construction of $\phi$ gives directly.

$$\phi_{\tilde{l},\tilde{i}}\left(\frac{i}{2^j}\right) = \frac{9}{16}\phi_{\tilde{l},\tilde{i}}\left(\frac{i+1}{2^j}\right) + \frac{9}{16}\phi_{\tilde{l},\tilde{i}}\left(\frac{i-1}{2^j}\right) - \frac{1}{16}\phi_{\tilde{l},\tilde{i}}\left(\frac{i+3}{2^j}\right) - \frac{1}{16}\phi_{\tilde{l},\tilde{i}}\left(\frac{i-3}{2^j}\right).$$

Therefore, we obtain $I_{l,i}\phi_{\tilde{l},\tilde{i}} = 0$. Finally, $I_{l,i}u = \sum_{\tilde{l},\tilde{i}} v_{\tilde{l},\tilde{i}} \cdot I_{l,i}\phi_{\tilde{l},\tilde{i}} = v_{l,i} \cdot I_{l,i}\phi_{l,i} = v_{l,i}$. This proves the stencil representation of $v_{\boldsymbol{l},\boldsymbol{i}}$ for dimension 1. Finally, using the tensor product approach of the stencil operator $I_{\boldsymbol{l},\boldsymbol{i}}$ we can generalize the formula to general dimensions. $\qquad\square$

