# OpenReview forum: "Shallow and Deep Networks are Near-Optimal Approximators of Korobov Functions"
_ICLR.cc/2022/Conference — ICLR 2022 Poster_

### Official Review · Reviewer_X13n · 2021-11-01

**Correctness:** 3
**Technical Novelty And Significance:** 2
**Empirical Novelty And Significance:** 2
**Recommendation:** 6
**Confidence:** 3

**Details Of Ethics Concerns:**

-

**Main Review:**

Strengths:
-in-depth study of Korobov function approximations
-several new ideas/gadgets on how to represent such functions with NNs


Weaknesses:
-unclear presentation in several places: how do the bounds break the curse of dimensionality as stated in bullet point 1?
-lack of motivation: Korobov functions seem like an interesting object for study, however this falls outside the community's "standard" knowledge, so the reviewer would expect much better explanation for why one should care for these functions in the context of Machine Learning or so. For example, the Conclusion states: "This work therefore contributes to understanding the practical success of neural networks theoretically" --> How is this the conclusion? Are Korobov functions prevalent in practice? The bounds in Th. 4.1 also seem to suggest that increasing depth from 2 to logd only shaves an sqrt{\eps} factor which does not seem so significant, even though the depth increased substantially.


Q: Why do the authors claim that they break the curse of dimensionality (e.g., page 2 bullet point 1?)? There is a strong dependence on d, right? What am I missing?

**Summary Of The Paper:**

This paper studies approximation capabilities of neural networks for the purpose of approximating Korobov functions which are multivariate functions of bounded second mixed derivatives.

The paper presents a complete study for approximating such functions with NNs: they study shallow nets and show that 2 layers with ReLUs and total #neurons of O(1/eps log^{1.5d}(1/eps))  where d is the dimension, can \eps-approximate  Korobov functions. Moreover, by allowing larger depths, close to logd, they can get a better depedence on \eps. Finally, they prove that any continuous function approximator requires a #params close to their upper bound in order to approximation Korobov functions.

This gives a complete picture for how Korobov functions behave wrt to function approximation with shallow or deep nets.



**Summary Of The Review:**

Except several relatively minor issues with presentation and clarification of the results, the reviewer thinks this paper to be a good paper on the topic of approximating Korobov functions. More work is needed for presentation of the paper, and fixing a few things here and there (see above as well)

---

> ### Author Response · Authors · 2021-11-17
> **Part 1/2: Breaking the curse of dimensionality and Korobov motivation**
>
> ___Part 1/2___
>
> We kindly thank the review for the detailed review. Below, we address the question and the raised concerns. We also suggest, if the reviewer thinks useful, to clarify more some of these concerns in the revision of the paper as precised below.
>
> **Question:** Why do the authors claim that they break the curse of dimensionality (e.g., page 2 bullet point 1?)? There is a strong dependence on d, right? What am I missing?
>
> **Answer:** Thank you for raising this subtlety. We claim to break the curse of dimensionality in the point of view asymptotic in $\epsilon$. In fact, asymptotically in $\epsilon$, $O((\log \epsilon^{-1})^{1.5(d-1)}) = o(\epsilon^{-\delta})$ for all $\delta>0$, hence our bound is $O(\epsilon^{-1-\delta})$ which breaks the curse of dimensionality in that sense. We note that this point of view, asymptotic in $\epsilon$, is well established in the numerical analysis literature [1] and in the neural network approximation power literature (e.g. see [2,3,4]).
>
> This is indeed a subtlety that requires good care in the presentation of the result. Throughout the paper, we made sure to precise what we mean by breaking the curse of dimensionality each time we mentioned it. When we do not specify that it is asymptotic in epsilon, we say "lessening the curse of dimensionality" instead, as the base exponent of $d$ in our bound is in $O(\log \epsilon^{-1})$ instead of the usual $O(\epsilon^{-1})$. With this point of view, this gain is also significant as other smooth spaces of functions such as Sobolov are proven to require $O(\epsilon^{-d})$ neurons for approximation.
>
> [1] *Novak, E. (2006). Deterministic and stochastic error bounds in numerical analysis.*
>
> [2] *Montanelli, H., \& Du, Q. (2019). New error bounds for deep ReLU networks using sparse grids. SIAM Journal on Mathematics of Data Science, 1(1), 78-92.*
>
> [3] *Mhaskar, H. N. (1996). Neural networks for optimal approximation of smooth and analytic functions. Neural computation, 8(1), 164-177.*
>
> [4] *Yarotsky, D. (2017). Error bounds for approximations with deep ReLU networks. Neural Networks, 94, 103-114.*
>
> **Question:** lack of motivation: Korobov functions seem like an interesting object for study, however this falls outside the community's "standard" knowledge, so the reviewer would expect much better explanation for why one should care for these functions in the context of Machine Learning or so. For example, the Conclusion states: "This work therefore contributes to understanding the practical success of neural networks theoretically" --> How is this the conclusion? Are Korobov functions prevalent in practice?
>
> **Answer:** We thank the reviewer for the instructive question. Korobov spaces present two motivations for the study of neural networks approximation power.
>
> The first motivation is theoretical. Very few previous works have been able to establish that neural networks can break the curse of dimensionality in approximating generic function spaces. Most results focus on specific function spaces.
>
> Perhaps the most natural way of defining a general space to study is by imposing smoothness conditions. This results in the space of Sobolev functions which was studied by Mhaskar [1] and others in the context of neural networks approximation power. However, DeVore showed that it was impossible to circumvent the curse of dimensionality in Sobolev spaces with any function approximator. Hence, a key question is what additional regularity on derivatives does the Soboloev space need to be able to be approximated without the curse of dimensionality and subsequently, if the resulting space can be approximated with neural networks. Korobov functions are Sobolev function with additional regularity on higher order derivatives (bounded mixed derivative instead of simple derivatives, see Appendix A). They arise naturally in the numerical analysis literature as the generic space that can be approximated without the curse of dimensionality with function approximators (see Appendix A for intuition). This makes the Korobov space a key candidate to analyze neural network approximation power and explore whether they break the curse of dimensionality for generic spaces of functions
>
> The second motivation arises from practice. Korobov spaces are useful for solving partial differential equations [1] and have been used in the context of high-dimensional function approximation [2]. They hence are an interesting function space to approximate with neural network when the objective is for instance to approximate solutions of partial differential equations.
>
> [1] *NM Korobov. Approximate solution of integral equations. Doklady Akademii Nauk SSSR, 128(2):
> 235–238, 1959.*
>
> [2] *Ch Zenger and W Hackbusch. Parallel algorithms for partial differential equations. In Notes on
> Numerical Fluid Mechanics, volume 31, pp. 241–251. Vieweg, 1991.*
>
> ___Continuity in the next comment___

---

> > ### Author Response · Authors · 2021-11-17
> > **Part 2/2: On Theorem 4.1 and benefit of depth**
> >
> > ___Part 2/2___
> >
> > **Question:** The bounds in Th. 4.1 also seem to suggest that increasing depth from $2$ to $\log d$ only shaves an $\sqrt{\epsilon}$ factor which does not seem so significant, even though the depth increased substantially.
> >
> > **Answer:** Thank you for the question. In each of our bounds, we present both the number of training parameters (neurons with non-fixed weight that needs to be trained) and the number of neurons. Indeed the number of training parameters does not change when we add depth. However, the number of neurons decreases by a factor $\sqrt{\epsilon}$ and matches the number of training parameters. This shows that deep neural networks are more efficient than shallow neural network in the sense that shallow networks need more "dead" neurons to reach the same approximation power as deep networks while essentially having the same number of parameters.
> >
> > Furthermore, asymptotically in $\epsilon$, the leading term of the bound in $\epsilon$ is $O(\epsilon^{-1})$. Hence, asymptotically in $\epsilon$, the factor $O(\epsilon^{-0.5})$ is significant.

---

> ### Comment · Reviewer_X13n · 2021-11-18
> **Thanks for response!**
>
> Thanks for the author's response. I hope the newer version will incorporate the answer and the comments included.

---

> > ### Author Response · Authors · 2021-11-22
> > **We have incorporated the answer and the comments**
> >
> >
> > We have incorporated the answer and the comments.
> > - **Breaking the curse of dimensionality:** In addition to specifying the asymptotic view in $\epsilon$ in page 2 bullet point 1, we have also more precisely discussed it in Section 2.1 (The curse of dimensionality)  under Section 1 (Preliminaries).
> > - **Motivation of Korobov:** We have added motivations of Korobov space in the 5th paragraph of the introduction and discussed one of them more in details in Section 2.1, last paragraph.
> > - **Benefit of depth and Thm 4.1:** We have added a discussion on the benefit of depth right after Theorem 4.1.
> >
> > Thank you again for the suggestions! The authors welcome any other suggestion to the paper in the next revision.

---

### Official Review · Reviewer_nqre · 2021-11-02

**Correctness:** 4
**Technical Novelty And Significance:** 3
**Empirical Novelty And Significance:** Not applicable
**Recommendation:** 6
**Confidence:** 3

**Main Review:**


#### Strengths
1. The paper gives matching upper bound and lower bound for the $X^{2, \infty}$ space.

#### Weakness
1. The space seems far away from the practice of deep learning, and the contribution is mainly theoretical.
    * The bound still has dependency on $(\log \epsilon^{-1})^{O(d)}$, which is still exponential. (Note that $\log \epsilon^{-1}$ is about the number of bits to represent a float point number with $\epsilon$ accuracy.) This makes it less motivated to study this function space.
    * The approximation issue is not the main concern in deep learning theory. Deep neural network merits not only because they are universal function approximators, but also, and more importantly, they can be efficiently trained and can generalize well. The optimization and generalization issues are the main concerns, for which this paper does not provide much insight.
    * This paper looks tangential to the community of ICLR. Maybe this paper is more suitable to venues in applied math or numerical analysis?
2. The technical novelty is rather limited, and the techniques looks to be rather limited to this specific problem of approximating Koborov space.
    * The upper bound construction seems to be based on a simple interpolation argument. It is strange why the previous paper (Montanelli & Du, 2019) did not achieve this bound. Maybe the authors could highlight the main technical challenges?
    * The techniques in this paper seem to be ad-hoc for the Korobov space and do not provide much insight for the
3. The authors should remove Appendices E-K when finalizing their paper.

**Summary Of The Paper:**

This paper studies what function class can be efficiently approximated by neural networks. This paper focuses on a special function class, namely the Korobov function space $X^{2, \infty}$, which contains function with $L^\infty$ bounded weak $\alpha$-order derivative, where $\| \alpha \|_\infty \le 2$. This is a local constraint on the local smoothness of the function space. This paper shows that shallow (with depth=$2$) and deep neural networks can efficiently approximate $X^{2, \infty}$, and the number of parameters does not scale with $\varepsilon^{-\mathrm{poly}(d)}$, thus efficiently escaping the curse of dimensionality. Furthermore, this paper shows the optimality of their parameters bound by showing a matching lower bound.

**Summary Of The Review:**


Because of concerns in the significance and technical novelty in this paper, I would recommend weak reject.

**After Rebuttal:** Thank the authors again for a detailed reply. The authors have addressed my main concerns. I would like to raise my recommendation to weak accept and increase the correctness and novelty scores, and confidence. I suggest the authors to incorporate their answers, especially 1.(a) and 2.(a), in their next revision.

---

> ### Author Response · Authors · 2021-11-17
> **Part 1/3: Answers to question and concerns on the relevance of the space and the results in deep learning**
>
> ___Part 1/3___
>
> We thank the reviewer for the detailed comments. Below we present answers to the raised questions and clarify some concerns.
>
> ___1. Relevance of the space and the contributions___
>
> **Concern:** 1(a) The bound still has dependency on $(\log \epsilon ^{-1})^{O(d)}$, which is still exponential. (Note that $(\log \epsilon ^{-1})$ is about the number of bits to represent a float point number with  accuracy.) This makes it less motivated to study this function space.
>
> **Answer:** This exponential dependence raises indeed a subtlety in the approximation literature.
> Our study is asymptotic in $\epsilon$ with fixed $d$. Under this setting, $(\log \epsilon ^{-1})^{d} = o(\epsilon^{-\delta})$ for all $\delta>0$. Hence the exponential dependence in $d$ is negligible in the bound. We note that this setting (asymptotic in $\epsilon$) is a well established in the neural network approximation power literature [1,2,3] and numerical analysis literature [4].
>
> Furthermore, one of Korobov space motivations in numerical analysis is precisely in the fact that it breaks the curse of dimensionality asymptotically in $\epsilon$ and lessens it in general (as the base exponent of $d$ becomes $\log \epsilon ^{-1}$) [5]. This gain is significant for approximating functions and cannot be observed with other generic smooth function spaces such as Sobolov, where DeVore [6] showed that one cannot escape a dependency in $\epsilon^{-d/r}$ where $r$ is the regularity.
>
> [1] *Montanelli, H., \& Du, Q. (2019). New error bounds for deep ReLU networks using sparse grids. SIAM Journal on Mathematics of Data Science, 1(1), 78-92.*
>
> [2] *Mhaskar, H. N. (1996). Neural networks for optimal approximation of smooth and analytic functions. Neural computation, 8(1), 164-177.*
>
> [3] *Yarotsky, D. (2017). Error bounds for approximations with deep ReLU networks. Neural Networks, 94, 103-114.*
>
> [4] *Novak, E. (2006). Deterministic and stochastic error bounds in numerical analysis.*
>
> [5] *Hans-Joachim Bungartz and Michael Griebel. Sparse grids. Acta numerica, 13(1):147–269, 2004.*
>
> [6] *Ronald A DeVore, Ralph Howard, and Charles Micchelli. Optimal nonlinear approximation. Manuscripta mathematica, 63(4):469–478, 1989.*
>
> **Concern:** 1(b) The approximation issue is not the main concern in deep learning theory. Deep neural network merits not only because they are universal function approximators, but also, and more importantly, they can be efficiently trained and can generalize well. The optimization and generalization issues are the main concerns, for which this paper does not provide much insight.
>
> **Answer:** The authors agree with the reviewer that training and generalization issues are main concerns in deep learning theory. Approximation power is closely linked to both these questions. In what follows we briefly outline the intuition behind this link.
>
>  Regarding generalization, the number of parameters that we study is the fundamental number of parameters neural networks need to approximate the given function (see Section 5 for a rigorous definition). This number is somehow the "fundamental" number of degrees of freedom needed to approximate this large class of functions with neural networks. Such a number is closely linked to generalization. If the class of approximators needs a large number of parameters, then it needs several "fundamental" degrees of freedom. More degrees of freedom imply more flexibility and hence poor generalization. On the contrary, if the class of approximators needs fewer degrees of freedom (we show neural network use the theoretically minimum number of degrees of freedom), then less degrees of freedom leads to better generalization.
>
> Regarding training, our results quantify the number of training parameters: the number of neurons that need to be trained. Less training parameters lead to easier training as we need to optimize a function with fewer variables when training the network. Therefore, our work suggests that neural networks require an optimal number of parameters to train to approximate a large class of smooth functions.
>
> We also point out that the literature on neural network approximation power is well established and these approximation power questions have been of interest to the community. We refer to the fourth paragraph of the introduction for some main references (we also added new references in the revision).
>
> ___Continuity in the next comment___

---

> > ### Author Response · Authors · 2021-11-17
> > **Part 2/3 Relevance to ICLR and Technical novelty**
> >
> > ___Part 2/3___
> >
> > **Question:** 1(c) This paper looks tangential to the community of ICLR. Maybe this paper is more suitable to venues in applied math or numerical analysis?
> >
> > **Answer:** Although this paper is of interest to applied math and numerical analysis communities, we believe the paper is also of strong interest to the ICLR community. In fact, several papers on neural network approximation power have been recently published in ICLR such as [1] in ICLR 2017 and [2] in ICLR 2018.
> >
> > [1] *Why deep neural networks for function approximation? Shiyu Liang, R. Srikant*
> > [2] *Adaptivity of deep ReLU network for learning in Besov and mixed smooth Besov spaces: optimal rate and curse of dimensionality. Taiji Suzuki.*
> >
> > ___2. Technical novelty and generalizability to other spaces___
> >
> > **Question:** 2(a) The upper bound construction seems to be based on a simple interpolation argument. It is strange why the previous paper (Montanelli & Du, 2019) did not achieve this bound. Maybe the authors could highlight the main technical challenges?
> >
> > **Answer:** The main technical challenges in our paper, different from (Montanelli \& Du 2019), are in shallow networks bounds and lower bounds results (which are our main contributions). For deep neural networks, our construction is similar to that of Montanelli \& Du, it uses however a more efficient approximation of the product which improves the approximation rates and reduces considerably the depth. Below we highlight some of main technical challenges and technical differences with Montanelli \& Du in our shallow networks bound and lower bound results.
> >
> > + *Shallow networks:* The common idea in our paper with Montanelli \& Du is the use of sparse grids to approximate smooth function with sums of products and then construct = neural networks to approximate this structure. One of the key difficulties in approximating this structure is to approximate the product function with neural networks efficiently. Montanelli \& Du use the previously known deep network architecture that approximates the product without the curse of dimensionality [1]. They use this architecture combined with sparse grids to get the desired rates of approximation. In the case of shallow neural networks, to the best of our knowledge, no architecture approximating the product function with a reasonable number of neurons was known before our work. We introduced the first shallow neural network to approximate the product (Proposition 3.2), which we combine with sparse grids to get our optimal approximation rates.
> >
> > + *Lower bound:* Our lower bound is new to the literature, and was not present in particular in Montanelli \& Du's work. In fact, classical existing lower bounds (for Sobolev space in particular) use $C^\infty$ function spaces to construct the lower bound using DeVore's theorem (Theorem 5.1). In the case of Korobov functions, DeVore's theorem requires the use of a linear space subset of Korobov functions with large Bernstein width (see end of page 8) which is a main challenge. We could construct such a space using Deslaurier-Dubuc interpolet of degree 2 [2].
> >
> > [1]  D. Yarotsky, Error bounds for approximations with deep ReLU networks, Neural Netw., 94 (2017), pp. 103--114.
> >
> > [2] Gilles Deslauriers and Serge Dubuc. Symmetric iterative interpolation processes. In Constructive
> > approximation, pp. 49–68. Springer, 1989.

---

> > > ### Author Response · Authors · 2021-11-17
> > > **Part 3/3: Generalizability of the techniques**
> > >
> > > ___Part 3/3___
> > >
> > > **Concern:** 2(b) The techniques in this paper seem to be ad-hoc for the Korobov space and do not provide much insight for the
> > >
> > > **Answer:** Our techniques can be extended to more general smooth function spaces. In fact, with the same architecture we can derive similar results for different spaces $X^{r,p}$ and norms $L_q$ for measuring the approximation error. The main dependence in $p,r,q$ results from the sparse grids bounds. We can therefore make extensions whenever similar results as Theorem 2.2 are verified. Bungartz and Griebel [1] show that these results generalize (see Theorem 3.8 and 4.8 in [1]). Using these results we can show the following bounds for our neural networks construction. We suggest, if the reviewer deems useful to introduce these extensions in our revision.
> > > + When $r=p=q=2$, we obtain the same bound $O(\epsilon^{-\frac{1}{2}}(\log \frac{1}{\epsilon})^{\frac{3}{2}(d-1)})$ as in the setting adopted in the paper ($r=2,p=q=\infty$).
> > > + In both spaces $r=2,p\in\{2,\infty\}$, we can obtain bounds for the energy norm and obtain $O(\epsilon^{-1})$ parameters.
> > > + For $r>2$, the resulting number of parameters is $O(\epsilon^{-\frac{1}{r}}(\log \frac{1}{\epsilon})^{\frac{r+1}{r}(d-1)})$ for both $p=q=\infty$ and $p=q=2$; and $O(\epsilon^{-\frac{1}{r-1}})$ for the energy norm with $p\in\{2,\infty\}$.
> > >
> > > Further, the techniques used to derive our lower bound on the number of parameters needed, are of potential interest for other spaces than the Korobov space. DeVore [2] shows that we can derive lower bounds from the maximum dimension of a ball of certain size that could be fitted within the function space (Theorem 5.1), which holds in a very general framework. For the special case of Korobov spaces, we showed that using a subspace inspired from the approximation space $V_n^{(1)}$ of Section 2 and 3---constructed to get \emph{upper} bounds---to bound the Bernstein width, yields an almost matching \emph{lower} bound. This observation might be true in more general settings as well.
> > >
> > > [1] *Hans-Joachim Bungartz and Michael Griebel. Sparse grids. Acta numerica, 2004.*
> > >
> > > [2] *Ronald  A  DeVore,  Ralph  Howard,  and  Charles  Micchelli. Optimal  nonlinear  approximation. Manuscripta mathematica, 63(4):469–478, 1989.*
> > >
> > > **Question:** The authors should remove Appendices E-K when finalizing their paper.
> > >
> > > **Answer:** We thank the reviewer for pointing this out. We missed it out when formatting the last version. We have fixed it in the revision.

---

> > > > ### Comment · Reviewer_nqre · 2021-11-21
> > > > **Thanks for the detailed reply!**
> > > >
> > > > I thank the authors for the detailed reply. The reply resolves most of my concerns.

---

> > > > > ### Author Response · Authors · 2021-11-22
> > > > > **We incorporated answers and comments in the revised manuscript**
> > > > >
> > > > > Thank you very much for your answer.
> > > > >
> > > > > We have incorporated the following discussions and comments into the new version of our manuscript.
> > > > > - *Lessening the curse of dimensionality:* In addition to specifying the asymptotic view in $\epsilon$ in page 2 bullet point 1, we discussed it more precisely in Section 2.1 (The curse of dimensionality)  under Section 1 (Preliminaries).
> > > > > - *Motivation of Korobov:* We added motivations of Korobov space in the 5th paragraph of the introduction and discussed one of them more in details in Section 2.1, last paragraph.
> > > > > - *Benefit of depth and Thm 4.1:* We added a discussion on the benefit of depth right after Theorem 4.1.
> > > > > - *Technical novelties:* We added an overview of the main technical challenges and novelties in the introduction (end of p2).
> > > > > - *Generalizability of our results:* We added a discussion of extensions of our upper bound constructions to more general Korobov spaces and in particular the spaces $X^{r,\infty}$ (end of Section 3).
> > > > >
> > > > > Thank you again for the suggestions! The authors welcome any other suggestion to the paper in the next revision.

---

### Official Review · Reviewer_bQiP · 2021-11-02

**Correctness:** 4
**Technical Novelty And Significance:** 3
**Empirical Novelty And Significance:** Not applicable
**Recommendation:** 8
**Confidence:** 3

**Main Review:**

I found paper to be very well written and easy to follow. The authors provide simple constructive proofs of the theorems, whose high-level ideas are well explained in the main text.

The discussion about the difference between Korobov and Sobolev spaces (Appendix A) is greatly appreciated. You mentioned there, that functions in Sobolev spaces can have more oscillations in various dimensions, and this can be the reason why they seem to be harder to approximate. It would be interesting to have a similar discussion to compare the Korobov space with the space of bandlimited function, which seem to avoid the curse of dimensionality. In particular, what do we intuitively gain in terms of expression power when extending the space of function from bandlimited functions to Korobov space ?

One question that would be interesting for practical purpose: By increasing the weights of the NN, one can approximate a function with any Korobov norm. In the proposed construction, what is the maximal weight magnitude that is used (in terms of dimension, accuracy and target function's Korobov norm) ? If the weight magnitude is restricted to a certain value, is it possible to design with a modified construction that account for such a constraint, and how does the number of required neurons/parameters would be affected. This question is of interest especially in scenarios where NNs are trained using weight decay or other regularisation.


Minor:
Pages 23-28 contain the formatting instruction which are of minor interest...
P.11 Sobolov
p.17 This proves the bound on the number of neutrons.

**Summary Of The Paper:**

This paper studies the ability of shallow and deep neural networks to approximate Korobov functions, and analyses their representation power in terms of the number of used parameters. The authors first show that 2 layers neural networks using common activation functions can approximate any Korobov function within eps error in infinity norm using O(eps^{-1/2} (log 1/eps)^{3(d-1)/2}) parameters. This result improves the existing upper bound of O(eps^{-d/r}) parameters for approximating Sobolev functions in W^{r,p} using 1-hidden layer neural networks, hence reducing the curse of dimensionality.

For deep neural network, this paper provides a new result, showing that neural networks with depth O(log d) and O(eps^{-1/2} (log 1/eps)^{3(d-1)/2}) approximate Korobov functions in X^{2, infinity} within eps error in L-infinity norm. This improves the result of [1], by requiring a depth independent of the required accuracy. However, the current work requires a C^2 and non-linear activation function, and is hence not applicable to ReLU.

Finally, the authors show that any continuous function approximator for the Korobov space requires O(eps^{-1/2} (log 1/eps)^{(d-1)/2}) parameters for achieving error eps, hence matching the previous upper bound for NNs up to factor (log 1/eps)^{d-1}.

[1] Hadrien Montanelli, Haizhao Yang, and Qiang Du. Deep ReLU networks overcome the curse of dimensionality for bandlimited functions

**Summary Of The Review:**

This paper is well written, and provides good theoretical insights about the representation power of both shallow and deep neural networks.

---

> ### Author Response · Authors · 2021-11-17
> **Author response [Part 1/2]: reducing the magnitude of weighs**
>
> Thank you for the positive review of our work and the detailed comments. We address below the questions raised.
>
> **Magnitude of weights:** Restricting the maximal weight magnitude is indeed an important practical concern. In the following, we show that the maximal weights in our construction can be made $O(1)$ independent of the dimension $d$, approximation error $\epsilon$ and the norm $|f|\_{2,\infty}$ by adding $O(\log \frac{|f|\_{2,\infty}}{\epsilon})$ new layers. Below we give a detailed explanation on how we can adapt our neural networks construction to account for maximal weights constraints. If the reviewer deems this discussion interesting to the reader, we could add it in the manuscript.
>
> In the proposed construction (Appendix B3), maximal weight amplitudes result from the weights of the first layer which approximates the 1-dimensional basis functions $\log \phi\_{l_j,i_j}$ (see p.17 on the revision) . Specifically, these are dilatations (and translations) of $\log \phi_{1,1}$ to the support $[\frac{i_j-1}{2^{l_j}},\frac{i_j+1}{2^{l_j}}]$. For the maximum value $l_j=n_\epsilon$ (see p. 18 on the revision), this accounts for a factor $2^{n_\epsilon}$ in the amplitude of the weights of the first layer. The maximal weight magnitude that is used in our construction is then $\frac{2^{n_\epsilon+1} |f|\_{2,\infty}}{\epsilon}\leq \frac{8}{8^{\frac{d}{2}}(2\log 2)^{\frac{d-1}{2}} (d-1)!^{\frac{1}{2}}}\left(\frac{|f|\_{2,\infty}}{\epsilon}\right)^{\frac{3}{2}} \left(\log \frac{1}{\epsilon} \right)^{\frac{d-1}{2}} \cdot (1+\mathcal O(1))$ --- the amplitude of weights on the next layers is significantly smaller: for the second layer the maximal weight is $\log \frac{|f|\_{2,\infty}}{\epsilon}+O(1)$ and the amplitude of weighs for the output node is at most $2^{-(d+2)}|f|\_{2,\infty}$.The factor $2^{n_\epsilon}$ resulting from dilating the basis function to a support of size $\frac{2}{2^{n_\epsilon}}$ can be circumvented by adding $n_\epsilon$ neurons after each input node and before the first layer of our initial construction, which each multiply the input by $2$. Therefore, by adding at most $d n_\epsilon \sim \frac{1}{2\log 2} d\log \frac{1}{\epsilon}$ neurons we can reduce the maximal weight amplitude of our network to $\frac{2 |f|\_{2,\infty}}{\epsilon}$ with no dependence on the dimension $d$.
>
> If we wish to remove the dependence in $\epsilon$ and $|f|\_{2,\infty}$, we can use the same idea by rescaling the first layer. Instead of approximating $\log \phi\_{l_j,i_j}$, we can approximate $(\epsilon/|f|\_{2,\infty}) \log \phi\_{l_j,i_j}$ then introduce $\log \frac{|f|\_{2,\infty}}{\epsilon}$ neurons which each multiply the output by at most $e$ before feeding the second layer of our initial construction. For each approximation of $\log \phi_{l_j,i_j}$ this adds $d 2^{n_\epsilon}\log \frac{|f|\_{2,\infty}}{\epsilon} = o(N)$ neurons where $N$ is the number of neurons in our original construction. The number of added layers is $\log \frac{|f|\_{2,\infty}}{\epsilon}$. With this new structure, the maximal weight amplitude of the first layer is $2$. We now turn to the "second layer" which approximates the exponential. Note that the only weights $\geq 1$ in this layer are due to the bias node (the exponential has to be approximated between $\log \frac{\epsilon}{6|f|\_{2,\infty}}$ and $0$). We can therefore introduce $\log \frac{6|f|\_{2,\infty}}{\epsilon}=o(N)$ neurons which each multiply by at most $e$ the value of the bias node. The maximal amplitude of this second layer is now $O(1)$. We finish our discussion with the output node. The maximal weighs of the output node in our original construction is $2^{(-d+2)}|f|\_{2,\infty}$ (see Theorem 2.2 and p.18). By adding $\log |f|\_{2,\infty}$ nodes after each approximation of the basis functions $\phi_{l,i}$, the maximal amplitude is now $O(1)$. The number of added layers in this step is $\log |f|\_{2,\infty}$ and we added $|U_n^{(1)}|\log |f|\_{2,\infty}=o(N)$ neurons. To conclude, the final modified neural network has $O(\log\frac{|f|\_{2,\infty}}{\epsilon})$ additional layers, $o(N)$ additional neurons where $N$ is the number of neurons in our original construction, and maximal weight amplitude $O(1)$, independent of $d$, $\epsilon$ and $|f|\_{2,\infty}$.
>
> From the fitting point of view, designing the structure of the neural network in order to remove the dependence in $|f|\_{2,\infty}$ requires the knowledge of the norm $|f|\_{2,\infty}$ beforehand or rather having an approximation of the norm up to a fixed multiplicative constant. These details were omitted in the above discussion.

---

> > ### Author Response · Authors · 2021-11-17
> > **Authors response [Part 2/2]: Comparison with bandlimited functions and minor comments**
> >
> > **Comparison with bandlimited functions:** Thank you for your instructive suggestion to add intuition of the difference between bandlimited functions and the Korobov space. We included such a discussion in appendix A of the revised manuscript. Although these spaces are different, we note that the bandlimited functions impose a hard constraint of "cutting" high frequencies by asking that the frequencies composing the function are restricted to a fixed compact. Instead, Korobov impose a weaker condition on frequencies by asking regularity. Intuitively, high frequencies are not prohibited but restrained by a budget as illustrated in the example end of Appendix A.
> >
> > **Minor comments:** Thank you very much for identifying the minor mistakes/typos in our previous document. We made the corresponding modifications in the revision of the manuscript.

---

### Official Review · Reviewer_G4AZ · 2021-11-02

**Correctness:** 4
**Technical Novelty And Significance:** 3
**Empirical Novelty And Significance:** Not applicable
**Recommendation:** 6
**Confidence:** 4

**Main Review:**

The paper is nicely written, and the theoretical results are well explained. I believe the results are correct and sound, albeit I did not check every detail in the proof.

I feel it should be good to discuss whether it is possible to consider higher-order mixed derivatives in Korobov spaces, for example, $X^{3, \infty}$. On the other hand, extending to bounded $L^p$ derivatives with $p < \infty$ can be interesting. Frankly speaking, I think the extension relies on if there are function approximation guarantees similar to Theorem 2.2 in the paper.

Compared to Sobolev spaces, we see the approximation of Korobov functions is free of the curse of data dimensionality. My impression is that there should be some low-dimensional structures in Korobov space allowing the circumvention. The authors explains why Korobov is easier to approximate in Appendix A, while I think some high-level remarks should be placed in Section 2.

The authors may want to discuss more related works on efficient approximation theories of neural networks. There are works considering either structures in function spaces, e.g., Besov functions with dominated mixed smoothness, or structures in data space, e.g., manifold data. These works all demonstrate that the size of neural networks in function approximation does not suffer from the curse of data dimensionality.

=========================================================================

Thanks for authors' detailed response. The generalization to $L_p$ norm guarantees is interesting. I am glad to hold a positive opinion on this paper.



**Summary Of The Paper:**

The paper proves upper and lower bounds on neural networks for approximating Korobov functions $X^{2, \infty}$. Upper and lower bounds match for approximation in $L^\infty$ norm sense, and the rate is free of the curse of data dimensionality. The scope of network architectures discussed is extensive, including shallow (2-layer) networks, deep networks with ReLU(-like) activation functions and Sigmoidal activation functions.

**Summary Of The Review:**

I am positive on the paper, due to its clarity and technical depth. I think the paper can be rated a 7, yet there is no rating between 6 and 8.

---

> ### Author Response · Authors · 2021-11-17
> **Generalization of results + Intuition on Korobov + Further literature overview**
>
> We are grateful to the reviewer for the very careful review and detailed comments. We address below the questions and suggestions.
>
> **Generalization to $X^{r,p}$:** Our paper focuses on approximating $X^{r,p}$ with $r=2$ and $p=\infty$. As noted by the reviewer, except for the bounds borrowed from the sparse grid literature, our approximation techniques are essentially independent from the choice of the space parameters $r,p$. In fact, while we used the infinity norm $L_{\infty}$ to evaluate the errors, we can also extend our bounds to other norms $L_q$. Below, we show how we can derive such extensions and present bounds for more general choices of $r,p$ and $q$.
>
> The dependency on $r$ and $p$ is essentially on the sparse grids part. More precisely, the number of training parameters and neurons corresponds essentially to the number of basis functions to be used for the sparse grid approximation $f^{(1)}_n$ (see Section 2.2). Thus, our bounds generalize to settings where similar results on the sparse grid construction are verified (Theorem 2.2). Bungartz and Griebel [1] show that these results indeed generalize (see Theorem 3.8 and 4.8 in [1]). Using these results we can show the following bounds for our neural networks construction:
> + When $r=p=q=2$, we obtain the same bound $O(\epsilon^{-\frac{1}{2}}(\log \frac{1}{\epsilon})^{\frac{3}{2}(d-1)})$ as in the setting adopted in the paper ($r=2,p=q=\infty$).
> + In both spaces $r=2,p\in\{2,\infty\}$, we can obtain bounds for the energy norm and obtain $O(\epsilon^{-1})$ parameters.
> + For $r>2$, the resulting number of parameters is $O(\epsilon^{-\frac{1}{r}}(\log \frac{1}{\epsilon})^{\frac{r+1}{r}(d-1)})$ for both $p=q=\infty$ and $p=q=2$; and $O(\epsilon^{-\frac{1}{r-1}})$ for the energy norm with $p\in\{2,\infty\}$.
>
> If the reviewer deems useful, we can include these extensions in the revision.
>
>
> **Intuitions on the Korobov space:** Thank you for your suggestion to give intuition on the Korobov space within Section 2. We added remarks in the revised paper to explain the benefits of the Korobov structure for function approximation in the first paragraph of Section 2, end of page 3.
>
> **Efficient approximation with neural networks:** The literature on function approximation with neural networks being very wide, we aimed to give a brief overview of the different lines of work in our manuscript. However, as suggested, we added more details in the literature review of our revised manuscript for the interested reader. In particular, we discussed the approximation function literature for compositions of functions, piece-wise smooth functions, approximation with data lying on manifolds, and results for the mixed Besov space [2].
>
> [1] *Hans-Joachim Bungartz and Michael Griebel. Sparse grids. Acta numerica, 2004.*
>
> [2] *Taiji Suzuki. Adaptivity of deep ReLU network for learning in Besov and mixed smooth Besov spaces: optimal rate and curse of dimensionality.arXiv preprint arXiv:1810.08033, 2018*

---

### Decision · Program_Chairs · 2022-01-20

**Decision:**

Accept (Poster)

**Comment:**

The authors theoretically analyze the approximation of Korobov
functions by neural networks, obtaining upper and nearly matching
lower bounds, for shallow and deep networks, with different activation
functions.  The bounds are stronger than what can be proved for the
more commonly studied Sobolev functions.  But Korobov functions are a
natural and fairly wide class of functions.  This work makes a
substantial step forward in clarifying what kind of functions are
especially amenable to representation by neural networks.  The
reviewers also appreciated the clarity of the writing.